# Rewiring of the phosphoproteome executes two meiotic divisions in budding yeast

Lori B Koch[1], Christos Spanos [1], Van Kelly[1], Tony Ly[1,2] & Adele L Marston [1✉]

## Abstract

**The cell cycle is ordered by a controlled network of kinases and phosphatases. To generate gametes via meiosis, two distinct and sequential chromosome segregation events occur without an intervening S phase. How canonical cell cycle controls are modified for meiosis is not well understood. Here, using highly synchronous budding yeast populations, we reveal how the global proteome and phosphoproteome change during the meiotic divisions. While protein abundance changes are limited to key cell cycle regulators, dynamic phosphorylation changes are pervasive. Our data indicate that two waves of cyclin-dependent kinase (Cdc28[Cdk1]) and Polo (Cdc5[Polo]) kinase activity drive successive meiotic divisions. These two distinct phases of phosphorylation are ensured by the meiosis-specific Spo13 protein, which rewires the phosphoproteome. Spo13 binds to Cdc5[Polo] to promote phosphorylation in meiosis I, particularly of substrates containing a variant of the canonical Cdc5[Polo] motif. Overall, our findings reveal that a master regulator of meiosis directs the activity of a kinase to change the phosphorylation landscape and elicit a developmental cascade.**

**Keywords** Meiosis; Cell Cycle; Proteomics; Phosphorylation; Kinases
**Subject Categories** Cell Cycle; Post-translational Modifications & Proteolysis; Proteomics

See also: D Celebic et al (February 2024)

## Introduction

In mitotically dividing cells, ploidy is maintained by strict alternation of DNA replication in the synthesis (S) phase and chromosome segregation in mitosis (M phase). The oscillating nature of the cell cycle is achieved through an inter-dependent combination of irreversible proteolytic degradation and reversible post-translational modifications. The major drivers of the cell cycle are cyclin-dependent kinases (Cdks). Cdk activity is low in G1, rises through S phase and peaks in mitosis after which Cdks are inactivated, triggering exit from mitosis and return to the G1 state. This state of low Cdk activity in G1 allows for the resetting of replication origins and entry into the next S phase. The events of mitotic exit are ordered by sequential inactivation of kinases, activation of phosphatases and selective dephosphorylation of particular substrates. In mammalian cells, proteolysis of cyclin B initiates mitotic exit, and thereafter the order of dephosphorylation is established by PP1 and PP2A phosphatase selectively dephosphorylating threonine over serine-centred sites (Holder et al, 2020; Kruse et al, 2020). In budding yeast, mitotic exit is ordered by sequential dephosphorylation of Cdk and Polo kinase phosphorylation sites and by the substrate preference of the phosphatase Cdc14 (Touati et al, 2018, 2019; Visintin et al, 1998).

The maintenance of ploidy in mitosis contrasts with meiosis where the goal is to generate gametes with half the ploidy of the parental cell. This is achieved by two consecutive chromosome segregation events (M phases), meiosis I and meiosis II, which follow a single S phase. Furthermore, meiosis I is a unique type of segregation event because homologues, rather than sister chromatids, are separated. This raises two key questions. First, how is the cell cycle machinery remodelled to direct the unique pattern of segregation in meiosis I, and second, what ensures that meiosis I is followed by another M phase, meiosis II, rather than S phase as in the canonical cell cycle?

The meiotic divisions are distinguished from mitotic division by three key features. First sister kinetochores attach to microtubules from the same, rather than opposite, poles in meiosis I (mono-orientation). Second, sister-chromatid cohesion is lost in two steps: from chromosome arms in meiosis I and from centromeres only in meiosis II. Third, there are two successive chromosome segregation events (two divisions). The Meikin family of meiosis-specific regulators (Spo13 in budding yeast, Moa1 in fission yeast, Matrimony in *Drosophila* and Meikin in mammals) are master regulators which establish these key events of meiosis (Galander and Marston, 2020). Although Meikin family proteins are poorly conserved at the sequence level, their common feature is the association with Polo kinase through its polo binding domain (PBD) and, where tested, this interaction is required for their meiotic functions (Matos et al, 2008; Galander et al, 2019; Kim et al, 2015; Bonner et al, 2020). This suggests that Meikin family proteins define the meiotic programme by influencing Polo-dependent phosphorylation. Indeed, Polo plays several essential functions in meiosis, though only some of them overlap with Meikin. For example, in budding yeast, depletion of Cdc5[Polo] causes a loss of monoorientation and arrest prior to meiotic division, while *spo13Δ* cells also lose monoorientation but undergo a meiotic division (Lee

[1]The Wellcome Centre for Cell Biology, Institute of Cell Biology, University of Edinburgh, Edinburgh EH9 3BF, UK. [2]Centre for Gene Regulation and Expression, School of Life Sciences, University of Dundee, Dundee DD1 5EH, UK. ✉E-mail: adele.marston@ed.ac.uk

and Amon, 2003; Lee et al, 2004; Clyne et al, 2003; Katis et al, 2004). Spo13 also influences the activity of other kinases to prevent meiosis II events, such as loss of centromeric cohesion and spore formation, occurring prematurely, though it is unclear if this is direct or through its association with Cdc5$^{Polo}$ (Oz et al, 2022; Galander et al, 2019).

In addition to meiosis-specific Polo regulation, the oscillations of Cdk activity are also expected to change to avoid completely exiting M phase after the first meiotic division. Indeed, early studies in the frog *Xenopus laevis* indicated that Cdk activity declines after anaphase I, but not completely, and that it quickly rises again during the transition to metaphase II (Furuno et al, 1994; Iwabuchi et al, 2000). More recently, analysis of sea star oocytes provided evidence that a rise in PP2A-B55 phosphatase activity at the meiosis I to II transition leads to the selective dephosphorylation of threonine-centred Cdk phosphorylation sites after meiosis I (Swartz et al, 2021). In budding yeast, progression through the meiotic divisions requires two major kinases, Cdc28$^{Cdk1}$ and a Cdk-related kinase Ime2 (Benjamin et al, 2003; Schindler and Winter, 2006). Distinct combinations of cyclins complex with Cdc28$^{Cdk1}$ during the meiotic divisions, however cyclin abundance does not correspond to cyclin-Cdk activity and cyclin post-translational modifications are likely to be important in defining cyclin-Cdk activity (Carlile and Amon, 2008). Both Cdk and Ime2 prevent loading of the replicative helicase at the meiosis I to II transition (Phizicky et al, 2018; Holt et al, 2007). The budding yeast Cdc14 phosphatase is critical for Cdk inactivation at mitotic exit (Visintin et al, 1998), but not at the meiosis I to II transition where it instead licences a second round of spindle pole body duplication (Marston et al, 2003; Bizzari and Marston, 2011; Fox et al, 2017). Together, these observations suggest that Cdk inactivation and phosphatase activation do occur at the meiosis I-meiosis II transition but only partially, achieving a kinase:phosphatase balance that maintains a subset of phosphorylations important for a subsequent M phase, rather than S phase, but how this is achieved is unclear.

To begin to understand the signalling events that create the meiotic programme and allow two sequential meiotic divisions, we have characterised the proteome and phosphoproteome of budding yeast cells synchronously undergoing the meiotic divisions. A combination of hierarchical clustering and motif analysis allowed us to infer the timing of activity of key meiotic kinases with implications for the control of meiosis. By generating a matched time-resolved dataset of *spo13Δ* cells, we reveal how a master regulator establishes the meiotic programme. Finally, by comparing the proteome and phosphoproteome of metaphase I-arrested wild-type, *spo13Δ* and *spo13-m2* cells, where Cdc5$^{Polo}$ binding is abolished, we identify substrates and a preferred motif targeted by Spo13-Cdc5$^{Polo}$ in meiosis I.

# Results

## A high-resolution proteome and phosphoproteome of the meiotic divisions

To discover the proteins and phosphorylation events at each stage of the meiotic divisions in budding yeast, we released cells from a prophase I block (Carlile and Amon, 2008) and confirmed synchrony by spindle morphology (Fig. 1A; Appendix Fig. S1A).

The total proteome and phosphoproteome were analysed by tandem mass tag (TMT) mass spectrometry at 15-min intervals spanning the meiotic divisions and prophase "time zero" (Fig. 1A; see "Methods"). In each of two biological replicates, we identified ~3600–4000 proteins with 3499 overlap between replicates (Fig. 1B). From singly phosphorylated peptides detected in at least two sequential timepoints, we quantified ~6700–8700 phospho-sites in each timecourse, with 4551 sites detected in both replicates (Appendix Fig. S1B). To more accurately analyse the dynamics of phosphorylation, we normalised the phospho-site abundances to their corresponding total protein abundance and quantified ~5500–7100 high-confidence sites (localisation score >0.75; (Olsen et al, 2006)) per replicate, of which 3877 were found in both replicates (Fig. 1B; Appendix Fig. S1B). The number of proteins and phospho-sites quantified in our experiments is comparable to other recent proteomic studies (Paulo et al, 2015; Cheng et al, 2018; Li et al, 2019). All subsequent analyses were restricted to the 3499 proteins and 3877 phospho-sites detected in both replicates, with phospho-sites normalised to their protein abundance (see "Methods" for details). The data for individual or groups of proteins can be visualised through an interactive web-based interface at https://meiphosprot.bio.ed.ac.uk/.

The numbers of protein and phospho-sites detected, and their distributions, were comparable at all timepoints (Appendix Fig. S1C,D). Slightly fewer phospho-sites were detected in prophase, indicating de novo phosphorylation thereafter (Appendix Fig. S1D). Serine-centred phospho-sites (85%) comprised the majority of the meiotic phosphoproteome with threonine (15%) and tyrosine (<1%) less abundant, comparable to mitosis (Touati et al, 2018) (Fig. 1C). We confirmed that our dataset has sufficient resolution to capture the expected dynamics of proteins and their phosphorylation across the divisions. Two waves of securin (Pds1) destruction were detected in anaphase I and II (Salah and Nasmyth, 2000) and the Spo20 prospore membrane formation protein increased in abundance, specifically during meiosis II (Neiman, 1998) (Fig. 1D). Similarly, phosphorylation of the nucleolar protein Net1 peaked in both anaphase I and anaphase II, when its inhibitory association with the cell cycle regulatory phosphatase Cdc14 is relieved (Marston et al, 2003; Buonomo et al, 2003), while phosphorylation of the meiosis-specific centrosome protein Spo74 peaked during meiosis II, when it carries out its role in prospore membrane formation (Nickas et al, 2003; Ubersax et al, 2003) (Fig. 1E). Together, these analyses confirm acquisition of a high-quality proteome and phosphoproteome of the meiotic divisions.

## Modest changes in global protein dynamics across the meiotic divisions

To identify the most dynamic meiotic proteins, we compared each of the nine timepoints spanning the divisions with prophase time zero. A median fold change in abundance greater than 1.5 and a *t* test value of $P < 0.05$ in at least one comparison defined 297 proteins as significantly dynamic, representing ~8% of the meiotic proteome (Appendix Fig. S2A; Dataset EV1). To reveal common dynamic trends across the timecourse in an unbiased manner, we used hierarchical clustering (Fig. 2A,B; Dataset EV2). Proteins were grouped into six clusters which each have a distinct pattern of abundance across the stages of meiosis. A group of proteins found specifically at prophase (cluster 3) was associated with metabolism

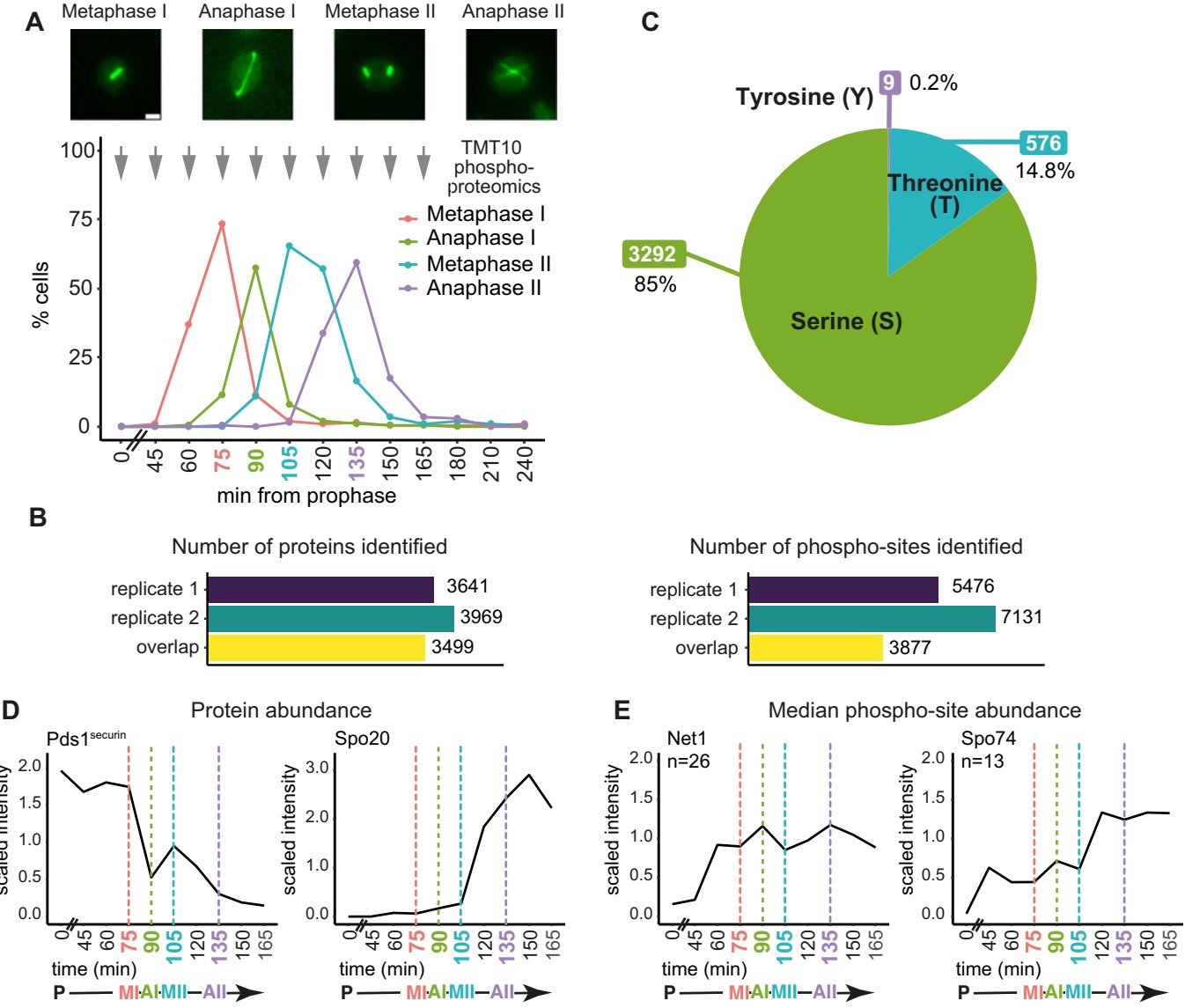

**Figure 1. Phosphoproteomics of a synchronous meiotic division cycle.**

(**A**) Example images and quantification of spindle immunofluorescence at the indicated timepoints after release of wild-type cells from a prophase I block. (n = 200 cells per timepoint). Arrows indicate time of harvesting for TMT10 proteomics and phosphoproteomics. Scale bar equals 2 μm. (**B**) Total number of proteins (left) and phospho-sites (right) quantified for each of two biological replicate wild-type TMT10 timecourses and those common to both timecourses (overlap). (**C**) The proportion of phospho-sites centred on serine, threonine or tyrosine. (**D**) Median protein abundance across the timecourse for selected proteins Pds1[securin] and Spo20. (**E**) Median abundance of all detected phospho-sites of the proteins Net1 and Spo74. n = number of phospho-sites. Source data are available online for this figure.

GO terms (Fig. 2C). Cluster 2 proteins were abundant from prophase until anaphase I and involved in chromosome pairing, cohesion, and recombination. Cluster 5 proteins were abundant in meiosis I and II and associated with meiotic cell cycle and sporulation GO terms. Clusters 1 and 4 had high abundance in meiosis II and included proteins involved in sister-chromatid cohesion/anaphase-promoting complex (APC) activity and spore wall assembly (Fig. 2C).

During the transition into anaphase I and II, the APC promotes degradation of key cell cycle regulators, including Pds1[securin] (Cohen-Fix et al, 1996; Salah and Nasmyth, 2000). Compared to their respective metaphase, 54 proteins were observed to

significantly change in anaphase I and 44 proteins in anaphase II, including a decrease in Pds1[securin] in both cases (Fig. EV1A,B). The cell cycle protein, Sgo1, and two APC regulators, Mnd2 and Acm1 (Penkner et al, 2005; Oelschlaegel et al, 2005; Enquist-Newman et al, 2008; Eshleman and Morgan, 2014; Clift et al, 2009) decreased in anaphase I, while the translational repressor Rim4, whose degradation promotes meiotic exit, decreased in anaphase II (Fig. EV1B; Berchowitz et al, 2013, 2015; Wang et al, 2020). Consistently, GO terms related to meiosis and the cell cycle were enriched among proteins that decreased in anaphase I and II (Fig. EV1C,D), while sporulation and cell wall proteins were among those that increased (Fig. EV1C,D). Therefore, despite representing

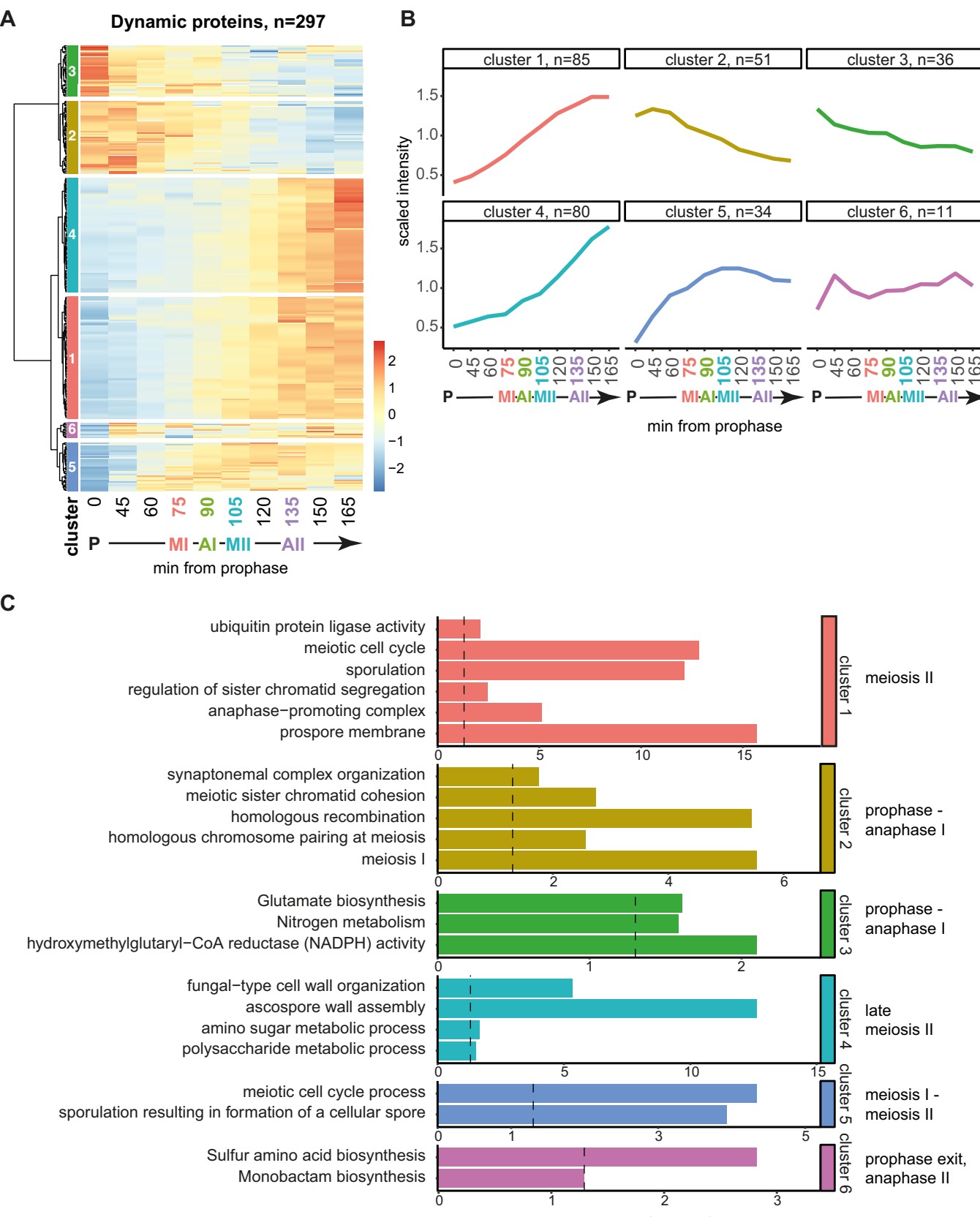

**Figure 2. Protein dynamics across a synchronous meiotic division cycle.**

(A) Hierarchical clustering of abundances of dynamic proteins across the timecourse. See Dataset EV2 for list of proteins included in each cluster. (B) Median trend of proteins in each cluster from (A). (C) GO term enrichment of proteins in each cluster from (A, B). Data information: Statistics: cumulative hypergeometric test followed by correction for multiple testing (gprofiler2 R package gost function default settings). Source data are available online for this figure.

only a small fraction of the proteome, dynamic proteins involved in key meiotic processes can be identified and new associations inferred by a combination of hierarchical clustering and GO term analysis. However, relatively few proteins change in abundance, indicating that other mechanisms must contribute to orchestrating the meiotic divisions.

## Distinct waves of phosphorylation across the meiotic divisions

To discover dynamic phosphorylation events during the meiotic divisions, we compared prophase to each of the other timepoints. We identified 1233 significantly dynamic phosphorylation sites of which 78% showed increased abundance after prophase, while ~21% decreased and 1% were variable depending on the timepoint (Appendix Fig. S2B; Dataset EV3). Hierarchical clustering produced 11 clusters (Fig. 3A) related by common biological processes (Fig. 3B). To determine whether these temporal patterns of phosphorylation are driven by particular kinases, we analysed the amino acid sequence surrounding the phospho-sites in each cluster (Appendix Fig. S3A; Dataset EV4). Over 50% of the phospho-sites in cluster 2, peaking in metaphase I and II, included a proline at the +1 position (Fig. 3C), which is the minimal consensus motif recognised by Cdc28$^{Cdk1}$, [ST]*P, suggesting bi-phasic activation of this kinase, consistent with previous studies (Carlile and Amon, 2008). Phosphorylation sites in cluster 3 were enriched for [DEN]x[ST]*, indicative of a peak of Cdc5$^{Polo}$-dependent phosphorylation in meiosis I (Fig. 3C).

We asked whether consensus phosphorylation motifs for 12 kinases expected to be involved in meiosis and identified in an in vitro peptide array analysis (Mok et al, 2010) were enriched in each of the time-resolved clusters (Appendix Fig. S3B,C, Fig. EV2; Table 1). Cdc28$^{Cdk1}$ and Cdc5$^{Polo}$ motifs were enriched in several clusters representing distinct waves of phosphorylation (Figs. 3D and EV2; Clusters 2–5 and 9). Cdc28$^{Cdk1}$ phosphorylation was enriched in prometaphase I (cluster 9), metaphase I and metaphase II (cluster 2) but significantly depleted from clusters with high phosphorylation in prophase (cluster 8), anaphase I (cluster 4 and 5) or anaphase II and late meiosis (clusters 5 and 6). Cdc5$^{Polo}$/Mps1 kinase-dependent phosphorylation appeared high throughout metaphase and anaphase of both meiotic divisions (clusters 3, 4, and 5). Prophase and prometaphase I clusters (Fig. EV2, clusters 8 and 9) were enriched for the Ipl1$^{Aurora B}$ consensus, consistent with its role in kinetochore disassembly and biorientation of homologous chromosomes, respectively (Miller et al, 2012; Chen et al, 2020; Meyer et al, 2013, 2015). Similarly, the meiosis-specific Mek1 kinase, which regulates inter-homologue recombination, appeared most active in prophase (Fig. EV2, clusters 8 and 11). The few sites matching the consensus motif for the meiosis-specific cell cycle kinase Ime2 (Appendix Fig. S3B,C) were enriched, though not significantly, at prophase exit (cluster 10) and in anaphase I (clusters 4, 5, and 10) (Fig. EV2).

Next, we grouped phospho-sites into categories based on their dynamics during the metaphase I-to-anaphase I or metaphase II-to-anaphase II transition. The majority of the 54 phospho-sites which decreased at anaphase I onset matched either the Cdc5$^{Polo}$ or Ipl1$^{AuroraB}$/PKA consensus, while around half of the 138 sites that increased matched either Cdc5$^{Polo}$, Ipl1$^{AuroraB}$/PKA or the minimal Cdc28$^{Cdk1}$ motif (Fig. EV3A). Thus, dynamic Cdc5$^{Polo}$-dependent phosphorylation occurs in meiosis I. At anaphase II onset, most of the 129 decreasing phospho-sites matched the minimal [ST]*P Cdc28$^{Cdk1}$ motif, suggesting that Cdc28$^{Cdk1}$ inactivation occurs at meiosis II, while no kinase motif was predominant among the 161 increasing sites (Fig. EV3B). In addition, we found no strong evidence that phosphorylation on motifs recognised by particular kinases disappear at different rates in either anaphase I or II (Fig. EV3C,D). The only exception was the strict Cdc28$^{Cdk1}$ consensus, [ST]*Px[KR], which declined slightly faster in both cases (Fig. EV3C,D). Furthermore, in contrast to mammalian mitotic exit and during the meiosis I to II transition in starfish (Swartz et al, 2021; Holder et al, 2020), threonine phosphorylation did not decline faster than serine phosphorylation (Appendix Fig. S4), suggesting no strong preference for dephosphorylation of threonine over serine at either anaphase, although subtle changes may be below the resolution of our dataset.

Two clusters of sites that were highly phosphorylated starting in meiosis II (clusters 1, 7) were not significantly enriched for any of the tested motifs (Fig. EV2) and more than half of the phospho-sites that increased after metaphase II did not match the consensus for any of the tested kinases (Fig. EV3B; "other"). Both cluster 1 (meiosis II-specific) phospho-sites and the phospho-sites that increased after metaphase II were enriched for acidic and phospho-acceptor residues in the surrounding sequence (Appendix Fig. S3A; Fig. EV3E), favoured by the casein kinases (Venerando et al, 2014; Mok et al, 2010). A potential candidate kinase for these sites is Hrr25$^{CK1}$, which increases in abundance in late meiosis (Fig. EV3F) and is important for spore formation (Argüello-Miranda et al, 2017).

Finally, phospho-proteins in clusters specific to meiosis I (clusters 3, 4, 9, 10), specific to meiosis II (clusters 5, 7) or with high phosphorylation during both divisions (cluster 2) showed little overlap and there was an enrichment of phospho-proteins uniquely present in meiosis II (Fig. 3E), suggesting that distinct proteins are phosphorylated during the two meiotic divisions.

This analysis identified the temporal order of phosphorylation during the meiotic divisions and predicts potential kinases responsible. Our data suggest signatures of Ipl1$^{Aurora B}$ phosphorylation mainly in prophase/prometaphase, distinct waves of Cdc28$^{Cdk1}$ and Cdc5$^{Polo}$ kinase phosphorylation in meiosis I and II and support the notion of upregulation of Hrr25$^{CK1}$ activity in meiosis II. We also identify patterns of phosphorylation and motifs where the kinase responsible is not easily predictable, which could indicate a role for uncharacterised kinases or altered specificity in key meiotic transitions.

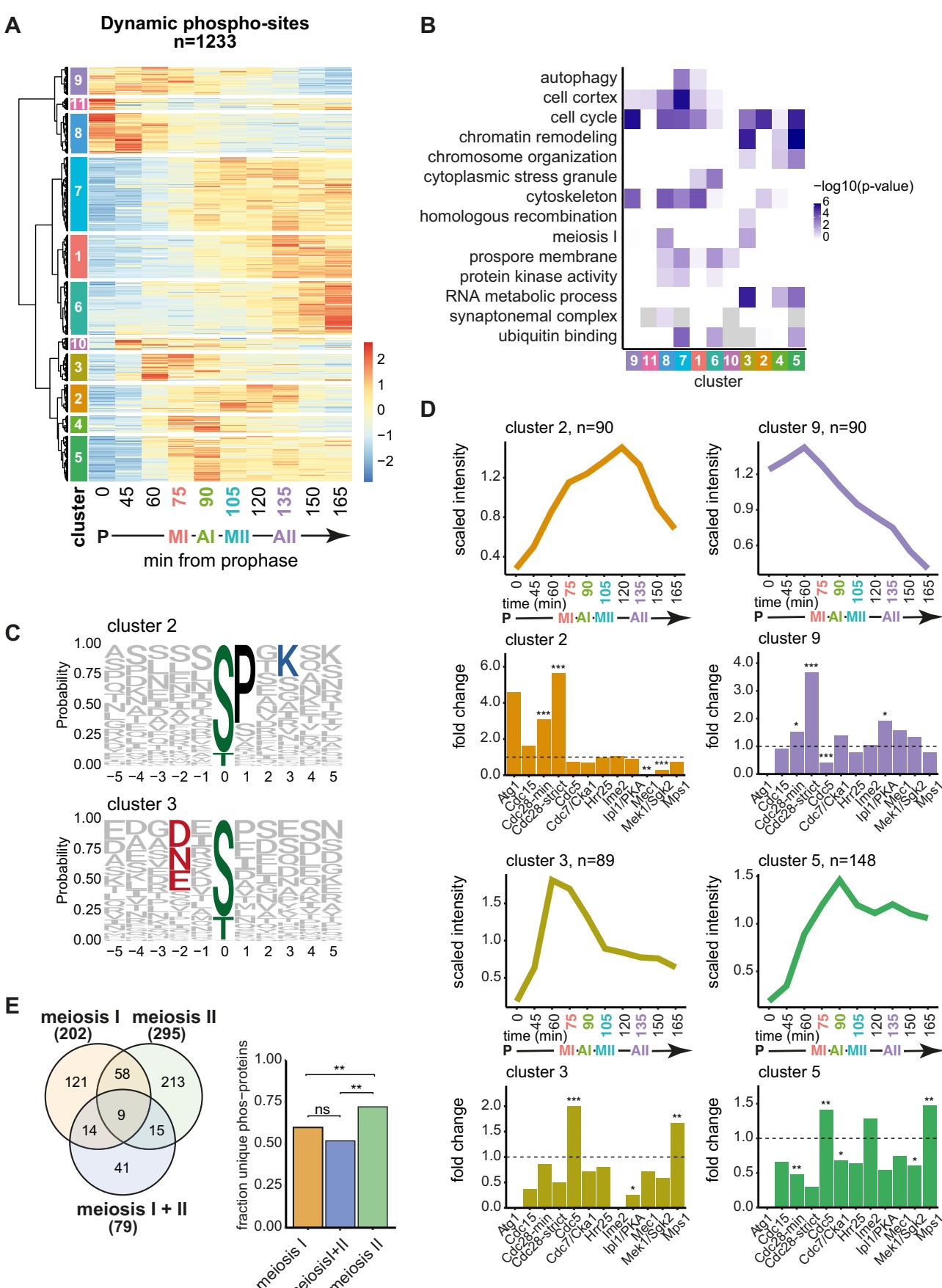

**Figure 3.** **The landscape of phosphorylation across the meiotic divisions.**

(A) Hierarchical clustering of dynamic phospho-sites. See Dataset EV4 for list of phospho-sites included in each cluster. (B) GO term enrichment of clusters. Data information: Statistics: Cumulative hypergeometric test followed by correction for multiple testing (gprofiler2 R package gost function default settings). (C) Motif logos of selected clusters. (D) Median lineplots of selected clusters with kinase motif enrichment analysis bar graph. Data information: Statistics: Fisher's exact test. *$P < 0.05$, **$P < 0.01$, ***$P < 0.001$. See Table 1 for motif list. (E) Venn diagram of meiosis I and meiosis II phospho-proteins grouped by clustering. Bar graph of fraction unique phospho-proteins in each group. Data information: Statistics: Fisher's exact test. **$P < 0.01$. Source data are available online for this figure.

**Table 1.** **Kinase consensus motifs used for analysis in Figs. 3D and EV2 and Appendix Fig. S3.**

| Kinase motif | Kinase name |
|---|---|
| [LM]xx[ST]*x[STVILMFYW][YIFM] | Atg1 |
| [ST]*x[RK] | Cdc15 |
| [ST]*P | Cdc28 minimal |
| [ST]*Px[KR] | Cdc28 strict |
| [DEN]x[ST]* | Cdc5 |
| [ST]*[DEST] | Cdc7/Cka1 |
| [ST]xx[ST]* | Hrr25 |
| RPx[ST]* | Ime2 |
| [RK]x[ST]* | Ipl1/PKA |
| [ST]*Q | Mec1 |
| [RK]xx[ST]* | Mek1/Sgk2 |
| [DER]x[ST]* | Mps1 |

## Proteome changes in spo13Δ cells are limited to a small number of key regulators

To determine how two distinct divisions are executed, we analysed the *spo13Δ* mutant in which the majority of cells undergo only a single meiotic division. We generated replicate proteome and phosphoproteome datasets of *spo13Δ* cells synchronously released from prophase as for wild type (Appendix Fig. S5). The number of proteins and phosphorylation sites identified and the overall distribution of total protein and phospho-site abundance was equivalent between strains and replicates (Appendix Figs. S5 and S6). For comparisons of wild type and *spo13Δ*, we restricted our analysis to proteins identified in at least one timepoint of all 4 experiments and to singly phosphorylated peptides that were detected in at least two sequential timepoints with high confidence (localisation score >0.75) of all 4 experiments. We identified 3296 proteins in both wild type and *spo13Δ* (Appendix Fig. S5C) and performed pairwise comparisons of protein abundances at matched timepoints to find proteins which varied by more than 1.5-fold reliably between replicates. This identified 50 proteins which differ in abundance between wild-type and *spo13Δ* at any one timepoint, corresponding to 1.5% of the shared wild-type-*spo13Δ* proteome (Appendix Fig. S7A; Dataset EV5). Differences included both increases and decreases in abundance and were observed at all stages of meiosis (Appendix Fig. S7B; Dataset EV6), consistent with the wide-ranging phenotypic effects of *spo13Δ*. Proteins that showed altered abundance in *spo13Δ* included key meiotic regulators Pds1[securin] and Sgo1 which showed reduced and delayed accumulation in *spo13Δ* cells (Appendix Fig. S7B,C; cluster 7). Some proteins involved in spore formation (Sps1, Sps22, Spr1) accumulated prematurely while the m6A methyltransferase

complex component, Vir1, which is required for the initiation of the meiotic programme (Park et al, 2023), was elevated in *spo13Δ* cells (Appendix Fig. S7B,C). Regulators Spo12 and Bud14 of the Cdc14 phosphatase, which is required for two meiotic divisions (Kocakaplan et al, 2021; Marston et al, 2003; Fox et al, 2017), were also deregulated in *spo13Δ* cells. Therefore, although overall protein dynamics in *spo13Δ* are similar to wild-type, the strict ordering of meiosis I and II events is lost for some key cell cycle and differentiation proteins. This implies that a combination of delayed meiosis I events and premature meiosis II events could explain the mixed single division of *spo13Δ* cells.

## Spo13 influences the dynamics of phosphorylation by major meiotic kinases

Spo13 controls the meiotic programme at least in part through its association with Cdc5[Polo], but the effect on global phosphorylation remains unknown. Comparison of phospho-site abundance between *spo13Δ* and wild type at each timepoint identified 305 phosphorylation sites whose abundance differed at one or more timepoints, representing ~13% of the shared phosphoproteome (Fig. 4A; Dataset EV7). Of these sites, the majority (203 or 67%) had reduced phosphorylation in *spo13Δ* versus wild type. To determine whether this could be explained by reduced phosphorylation by a particular kinase, we used IceLogo (Colaert et al, 2009) to compare the 203 sites with reduced phosphorylation in *spo13Δ* to those with increased phosphorylation or no change and found that asparagine (N) or glutamic acid (E) in the −2 position was overrepresented among decreased sites (Fig. 4B). This matches the canonical Polo kinase recognition motif, [DEN]x[ST]*, and the depletion of phosphorylation on these sites in *spo13Δ* suggests that Spo13 promotes Cdc5[Polo]-dependent phosphorylation on at least a subset of its meiotic targets. The group of sites with increased phosphorylation in *spo13Δ* (86 or 28%; Fig. 4A) showed enrichment for arginine (R) in the −3 position, matching the consensus for Ipl1[Aurora B], among other basophilic kinases (Fig. 4C). This likely indicates repeated rounds of Ipl1[AuroraB]-mediated error correction to reorient incorrect kinetochore–microtubule interactions in *spo13Δ* cells due to the absence of monopolin (Katis et al, 2004; Lee et al, 2004; Monje-Casas et al, 2007; Nerusheva et al, 2014).

Hierarchical clustering identified several groups of phospho-sites where the temporal trends across the meiotic divisions were altered in *spo13Δ* (Fig. 5A,B; Dataset EV8). Cluster 10 showed a peak of phosphorylation in meiosis I in wild type and a delay in phosphorylation in *spo13Δ* and was enriched for the strict Cdc28[Cdk1] motif [ST]*Px[KR] (Fig. 5B,C). Interestingly, this cluster included phosphorylation of S36 on the spindle pole body component Spc110 (Fig. 5C), which promotes timely mitotic exit (Abbasi et al, 2022), and we speculate could contribute to the single division phenotype of *spo13Δ* cells. Cluster 8, which peaks in

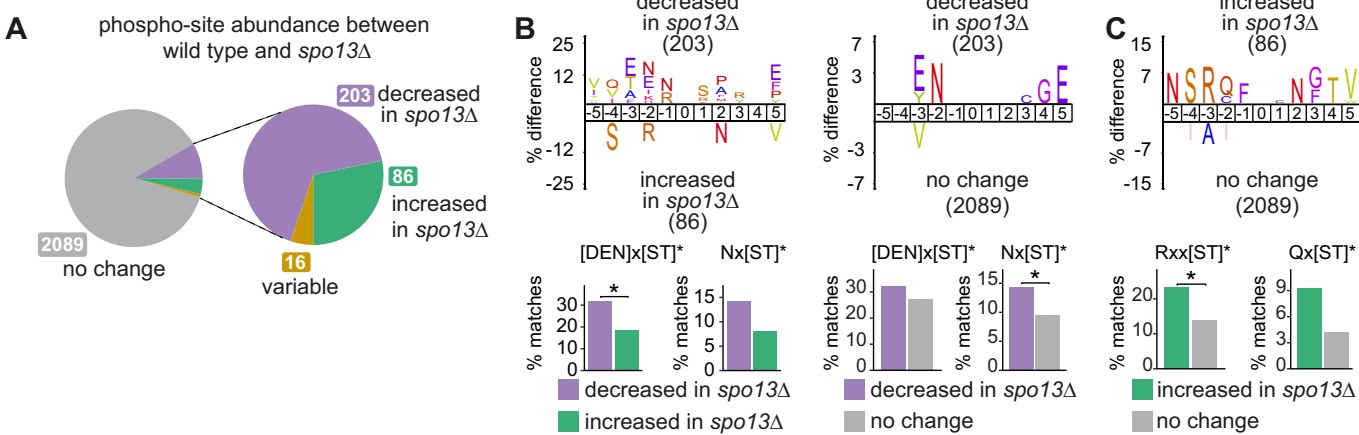

**Figure 4. Cdc5^Polo kinase motif phosphorylation is decreased during the meiotic divisions in spo13Δ cells.**

(A) Proportion of phospho-sites which significantly vary between wild type and spo13Δ at matched timepoints. See Dataset EV7 for list of phospho-sites. (B) IceLogos showing enrichment of specific residues surrounding the phospho-sites decreased in spo13Δ. Bar charts below show percent motif matches in the indicated groups of sites. Data information: Statistics: Fisher's exact test. *P < 0.05. (C) IceLogo showing enrichment of specific residues surrounding the phospho-sites increased in spo13Δ. Bar chart below shows percent motif matches in the indicated groups of sites. Data information: Statistics: Fisher's exact test. *P < 0.05. Source data are available online for this figure.

metaphase I-anaphase I in wild type, but not spo13Δ, is strongly enriched for the Cdc5^Polo consensus motif (Fig. 5B,D). Among these is a site on the synaptonemal complex component Ecm11 (S169; Fig. 5D), which could potentially be a relevant target for Cdc5^Polo-mediated synaptonemal complex disassembly (Argunhan et al, 2017). Cluster 3 is characterised by a sharp rise in phosphorylation after meiosis II which is absent in spo13Δ and is enriched for acidic residues typically targeted by Hrr25^CK1 (Fig. 5B,E). This further strengthens the idea that Hrr25^CK1-dependent phosphorylation is prevalent in late meiosis II (Fig. EV3E,F) and is consistent with a lack of coordinated Hrr25^CK1 activity in spo13Δ cells (Galander et al, 2019). Cluster 3 is exemplified by Cdc3-S77, a site within a component of the septin cytoskeleton that is required for efficient spore formation (Heasley and McMurray, 2016), a key function of Hrr25^CK1 (Argüello-Miranda et al, 2017). Finally, we note acidic residues at −3 and −2 of the cluster 9 consensus, suggesting potential phosphorylation by Cdc5^Polo or Hrr25^CK1 (Fig. 5F). Phosphorylation of Hrr25^CK1 itself on serine 330, which has been documented in mitotic cells (Breitkreutz et al, 2010; Zhou et al, 2021) was found in this cluster (Fig. 5F). Cdc5^Polo and Hrr25^CK1 physically interact (Galander et al, 2019), raising the possibility that Hrr25-S330 is regulated by Cdc5^Polo bound to Spo13.

To ask whether these effects of spo13Δ on the dynamics of phosphorylation by Cdc28^Cdk1, Cdc5^Polo, and Hrr25^CK1 were observable in the broader dataset, we plotted the mean-scaled abundance of sites matching the consensus for each kinase across the timecourse. The minimal Cdc28^Cdk1 consensus increased in abundance from prometaphase I in wild type, declining only after anaphase II, while the strict Cdc28^Cdk1 consensus revealed a clear bimodal pattern (Fig. 5G). In spo13Δ, both the minimal and strict Cdc28^Cdk1 consensus accumulated with a delay and decreased prematurely (Fig. 5G). The absence of a bimodal pattern of strict Cdc28^Cdk1 motif phosphorylation in spo13Δ is particularly striking (Fig. 5G). The consensus motifs for both Cdc5^Polo and Hrr25^CK1 peaked in prometaphase I (60 min), anaphase I (90 min) and

anaphase II (105 min) in wild type, but clear peaks were absent in spo13Δ (Fig. 5H,I). These findings confirm that meiosis I and II are characterised by distinct patterns of phosphorylation and that Spo13 promotes this ordering.

In summary, our data suggest that Spo13 promotes phosphorylation of a group of substrates by Cdc5^Polo kinase specifically in meiosis I and may affect Hrr25^CK1 function in late meiosis II. In addition, we find that the presence of Spo13 promotes two consecutive peaks of Cdc28^Cdk1 activity.

## [ST]*Px[KR] phosphorylation is the best predictor of Cdc28^Cdk1 kinase activity

Two distinct peaks of phosphorylation of [ST]*Px[KR] motif sites suggested two waves of Cdc28^Cdk1 activity in meiosis (Fig. 5G). To confirm that phosphorylation of this motif, and the minimal [ST]*P motif, can be attributed to Cdc28^Cdk1 in meiosis, we analysed the phosphoproteome after Cdc28^Cdk1 inhibition. We used cells where CDC28 is mutated such that the kinase can be specifically inhibited by the addition of the ATP analogue 1NM-PP1 (cdc28-as) (Bishop et al, 2000). To identify "meiosis I" phosphorylations 1NM-PP1 was added at prophase and samples were collected 75 min later (metaphase I) (Appendix Fig. S8A). For "meiosis II" phosphorylations, 1NM-PP1 was added during anaphase I (90 min after release from prophase I) and samples collected 15–30 min later, at the peak of metaphase II enrichment as judged by spindle morphology (Appendix Fig. S8A). We quantified ~700–800 high-confidence phospho-sites on ~3500 proteins (Appendix Fig. S9A). Cdc28-as inhibition reduced the number of [ST]*P sites detected, although it was only statistically significant in the meiosis II samples, and there was a significant reduction in the number of [ST]*Px[KR] sites detected at both stages (Fig. EV4A; Dataset EV9). We categorised phospho-sites detected in both wild-type and cdc28-as into those that decrease, increase or do not change upon Cdc28^Cdk1 inhibition (Fig. EV4B) and compared the surrounding amino acid sequences

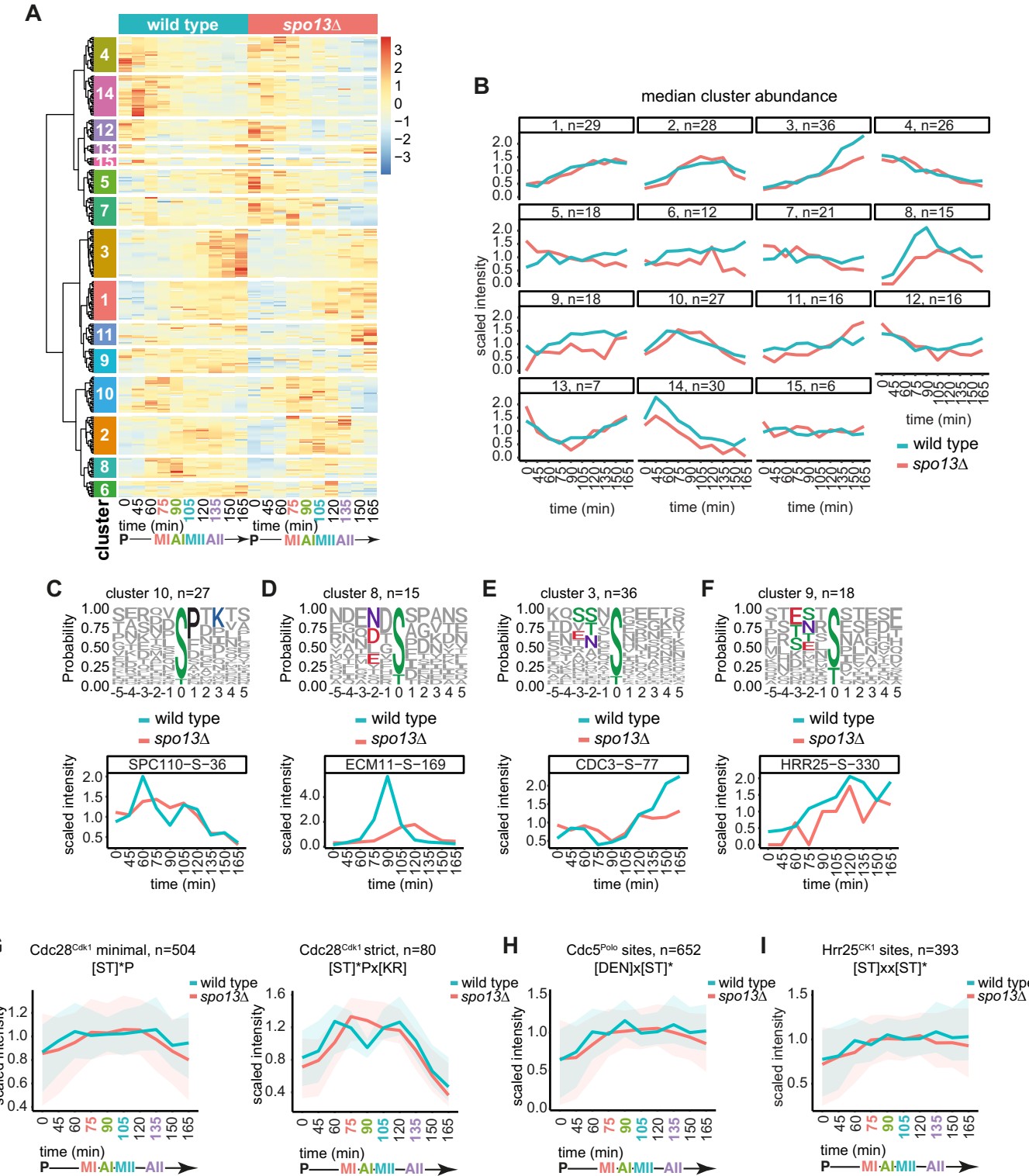

**A** wild type / spo13Δ — heatmap with clusters 4, 14, 12, 13, 15, 5, 7, 3, 1, 11, 9, 10, 2, 8, 6

**B** median cluster abundance

**C** cluster 10, n=27 — SPC110-S-36

**D** cluster 8, n=15 — ECM11-S-169

**E** cluster 3, n=36 — CDC3-S-77

**F** cluster 9, n=18 — HRR25-S-330

**G** Cdc28Cdk1 minimal, n=504 [ST]*P; Cdc28Cdk1 strict, n=80 [ST]*Px[KR]

**H** Cdc5Polo sites, n=652 [DEN]x[ST]*

**I** Hrr25CK1 sites, n=393 [ST]xx[ST]*

between the groups with decreased phosphorylation to those which do not change (Fig. EV4C–F). At both meiosis I and II, [ST]*Px[KR] was significantly enriched among sites decreased upon Cdc28-as inhibition, while [ST]*P was significantly enriched only at meiosis II (Fig. EV4C–F). In contrast, we found no enrichment

for the [DEN]x[ST]* Cdc5Polo motif among sites decreased in *cdc28-as* (Fig. EV4D,F), confirming that these sites are unlikely to be phosphorylated by Cdc28Cdk1. We conclude that Cdc28Cdk1 activity promotes phosphorylation of both [ST]*P and [ST]*Px[KR] sites during meiosis. Our data further suggest that [ST]*Px[KR]

◀  **Figure 5.  Cdc28<sup>Cdk1</sup>, Cdc5<sup>Polo</sup>, and Hrr25<sup>CK1</sup> phosphorylation is disrupted in *spo13Δ* cells.**

(A) Hierarchical clustering of the 305 phospho-sites which significantly vary between wild type and *spo13Δ* at matched timepoints. See Dataset EV8 for phospho-site identities in each cluster. (B) Median lineplots of cluster abundances from (A). (C) Motif logo of cluster 10 contains Cdc28<sup>Cdk1</sup> consensus, [ST]*Px[KR], matches (top). Abundance of S*PxK site Spc110-S36 phosphorylation across the timecourse (bottom). (D) Motif logo of cluster 8 contains Cdc5<sup>Polo</sup> consensus, [DEN]x[ST]*, matches (top). Abundance of NxS* site Ecm11-S169 phosphorylation across the timecourse (bottom). (E) Motif logo of cluster 3 contains Hrr25<sup>CK1</sup> consensus, [ST]xx[ST]* matches as well as acidic or phospho-acceptor residues at −2, +2, and +3 that are recognised by casein kinases (top). Abundance of SSxS* site Cdc3-S77 phosphorylation across the timecourse (bottom). (F) Motif logo of cluster 9 contains Hrr25<sup>CK1</sup> consensus, [ST]xx[ST]* and Cdc5<sup>Polo</sup> consensus, [DEN]x[ST]* matches (top). Abundance of DNxS* site Hrr25-S330 phosphorylation across the timecourse (bottom). (G) Abundance of phospho-sites matching the Cdc28<sup>Cdk1</sup> minimal [ST]*P (left) or strict [ST]*Px[KR] (right) consensus among sites detected in both replicates of wild-type and *spo13Δ* across the timecourse. (H) Abundance of phospho-sites matching the Cdc5<sup>Polo</sup>, [DEN]x[ST]*, consensus among sites detected in both replicates of wild-type and *spo13Δ* across the timecourse. (I) Abundance of phospho-sites matching the Hrr25<sup>CK1</sup>, [ST]xx[ST]*, consensus among sites detected in both replicates of wild-type and *spo13Δ* across the timecourse. Source data are available online for this figure.

phosphorylation is a better predictor of Cdc28<sup>Cdk1</sup> activity than [ST]*P phosphorylation and indicate that a subset of [ST]*P phosphorylation may be attributed to other kinases that recognise this motif e.g. MAP kinases; (Mok et al, 2010), particularly in meiosis I. Taken together with the observed bimodal phosphorylation of [ST]*Px[KR] sites in wild type, but not *spo13Δ* meiosis (Fig. 5G), this provides strong evidence for two distinct peaks of Cdc28<sup>Cdk1</sup> activity during the meiotic divisions and further suggests that Spo13 may contribute to bimodal Cdc28<sup>Cdk1</sup> activation.

## Cdc5<sup>Polo</sup> interacts with the cyclin Clb1 at metaphase I

To gain insight into how Spo13-Cdc5<sup>Polo</sup> might regulate the phosphoproteome more directly, including potential interactions with other kinases, we analysed Cdc5<sup>Polo</sup> immunoprecipitates from wild-type and *spo13Δ* metaphase I cells by mass spectrometry (Appendix Fig. S8B). As expected, Cdc5<sup>Polo</sup> associated with Spo13, Dbf4-dependent kinase (DDK), casein kinase Hrr25<sup>CK1</sup>, and components of the anaphase-promoting complex (APC) in wild-type cells (Fig. 6A; (Matos et al, 2008; Rojas et al, 2023; Galander et al, 2019; Chen and Weinreich, 2010), and associations with APC, DDK, and Hrr25<sup>CK1</sup> were maintained in the absence of Spo13 (Fig. 6B). Interestingly, the cyclin Clb1 also interacted with Cdc5<sup>Polo</sup>, in a Spo13-independent manner (Fig. 6B,C). Whether the Clb1-Cdc5<sup>Polo</sup> interaction is direct, or through the recently reported association of Clb1 with the APC activator Ama1 is unclear (Rojas et al, 2023). Nevertheless, Clb1 in Cdc5<sup>Polo</sup> immunoprecipitates showed reduced phosphorylation on several sites in *spo13Δ* (Fig. 6D,E), as was also observed upon direct immunoprecipitation of Clb1, where the phosphorylation could be attributed to Cdc5<sup>Polo</sup> activity (Rojas et al, 2023). Together with our finding that strict Cdc28<sup>Cdk1</sup> motif phosphorylation dynamics are significantly different in *spo13Δ* versus wild type (Fig. 5G), these data collectively raise the possibility that Cdc28<sup>Cdk1</sup> activity is regulated through Spo13-Cdc5<sup>Polo</sup>-dependent phosphorylation of Clb1. However, it is clear that other mechanisms are important in regulating Cdc28<sup>Cdk1</sup> since a Clb1 phosphonull mutant undergoes two divisions (Rojas et al, 2023) and other cyclins are known to be important (Dahmann and Futcher, 1995).

## Spo13 promotes Cdc28<sup>Cdk1</sup> and Cdc5<sup>Polo</sup> consensus phosphorylation at metaphase I

Spo13 is expected to exert its key function in metaphase I prior to its degradation in anaphase I (Sullivan and Morgan, 2007). To gain deeper insight into the effect of Spo13 and to understand the

importance of Cdc5<sup>Polo</sup> binding, we analysed the phosphoproteome of metaphase I-arrested wild-type, *spo13Δ* and *spo13-m2* cells. *spo13-m2* carries mutations in the motif that binds the Polo Binding Domain (PBD) of Cdc5<sup>Polo</sup> (Matos et al, 2008). Metaphase I arrest was confirmed and 3960 proteins and 6927 phosphorylation sites were identified in at least three replicates of wild-type, *spo13Δ* and *spo13-m2* strains (Appendix Fig. S10).

Between wild type and *spo13Δ* or wild type and *spo13-m2*, only a minor fraction of proteins 1.9% (115) or 0.9% (34), respectively, showed significantly different abundance at metaphase I (Fig. 7A,B; Datasets EV10 and EV11). Consistent with the timecourse, the mitotic exit network factor Spo12 was less abundant in *spo13Δ* at metaphase I (Appendix Figs. S7B,C and S11A). The m6A methyltransferase complex (MIS) complex protein Vir1 was also significantly changed in *spo13Δ* (Appendix Figs. S7B,C and S11A). This suggests that Spo13 has effects on protein abundance prior to anaphase I and raises the interesting possibility that Spo13 regulates gene expression. We note that *spo13-m2* has a more modest effect on protein abundance than *spo13Δ* (Fig. 7A,B; Appendix Fig. S11B), suggesting that Spo13 may affect protein abundance independently of Cdc5<sup>Polo</sup> binding. Indeed, Cdc5<sup>Polo</sup> accumulates only after prophase I exit (Sourirajan and Lichten, 2008), so any prior Spo13 functions would be expected to be mediated independently. However, an important caveat is that Spo13-m2 retains residual Cdc5<sup>Polo</sup> binding (Matos et al, 2008) which could account for its more modest effects.

Compared to wild-type at metaphase I, 893 and 272 phospho-sites were significantly different in *spo13Δ* and *spo13-m2*, respectively (Fig. 7A,B; Datasets EV12 and EV13). As in the timecourse experiments (Fig. 4A), the dominant trend was for reduced phosphorylation in *spo13Δ* compared to wild-type (626/893 or 70% of significantly different sites). A more modest effect was observed for *spo13-m2*, however a similar fraction of sites had reduced phosphorylation (211/272 or 78%). Motif analysis of sites with reduced phosphorylation in *spo13Δ* or *spo13-m2* versus wild type revealed an enrichment for aspartic acid or asparagine at −2 and for proline at +1, consistent with reduced Cdc5<sup>Polo</sup> and Cdc28<sup>Cdk1</sup> phosphorylation in the absence of functional Spo13 (Fig. 7C,D). In addition, we noted that fewer basophilic motif sites ([RK]x[ST]* or [RK]xx[ST]*), phosphorylation of which may be at least partially attributed to Ipl1<sup>AuroraB</sup>, had decreased phosphorylation in *spo13* mutants than expected (Fig. 7C,D), and there was also no increased phosphorylation of these sites in the mutants (Appendix Fig. S11C,D), in contrast with our results from the timecourse (Fig. 4C). This suggests that, for the most part, the differential phosphorylation of basophilic motif sites in *spo13Δ* does not occur at metaphase I.

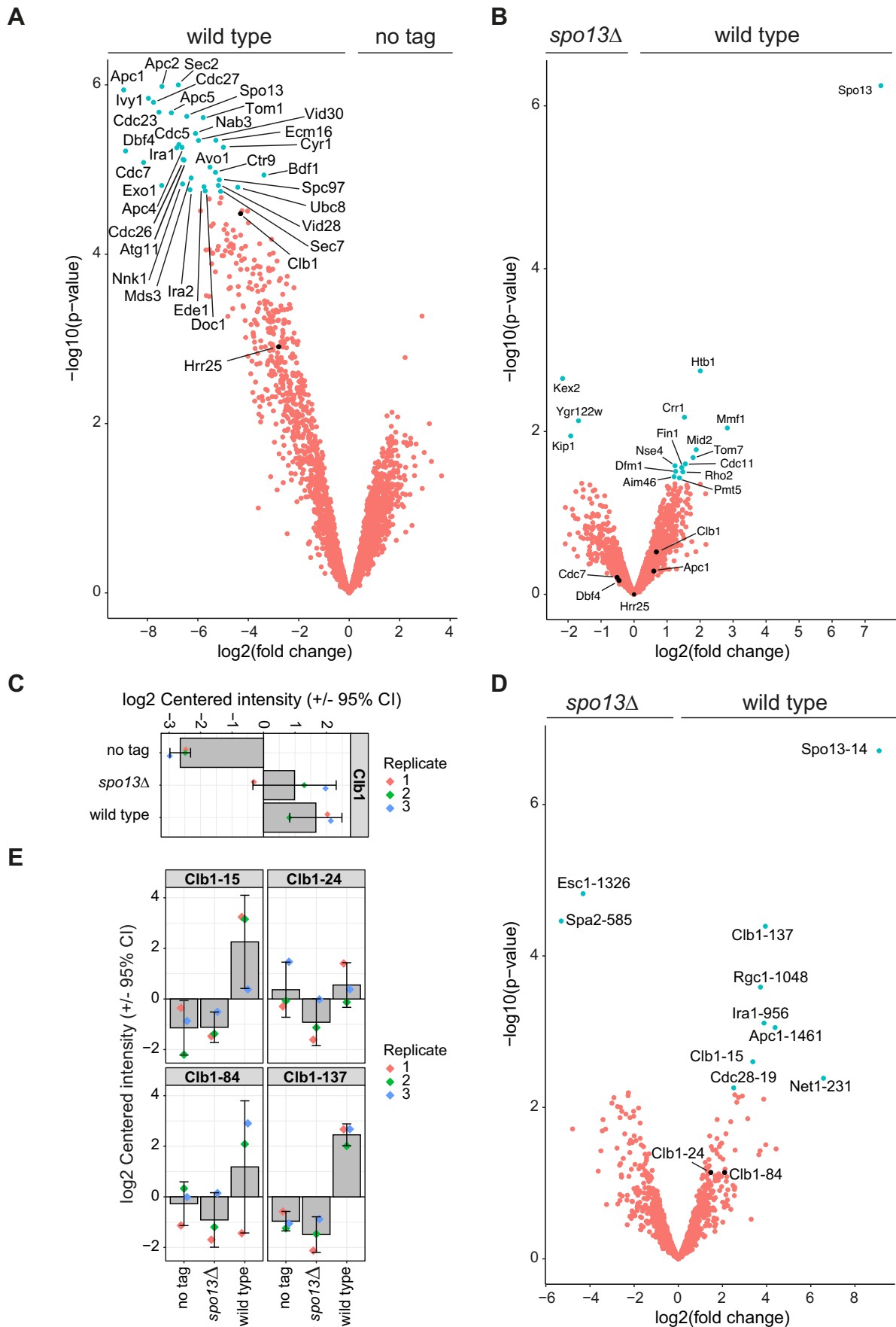

◄ **Figure 6. Cyclin Clb1 interacts with Cdc5$^{Polo}$ and its phosphorylation depends on Spo13.**

(A) Volcano plot comparing proteins that co-immunoprecipitate with Cdc5-V5 ("wild type") vs no tag control. (B) Volcano plot comparing proteins co-immunoprecipitated with Cdc5-V5 in wild-type and *spo13Δ* strains. (C) Abundance of Clb1 protein co-immunoprecipitated with anti-V5 in no tag, Cdc5-V5 ("wild type"), and *spo13Δ* Cdc5-V5 strains. Values are log2 transformed and mean-centred by protein, calculated by subtracting the mean abundance of the given protein across all samples from the abundance in a given sample. Error bars represent the 95% confidence interval around the mean. (D) Volcano plot comparing phospho-sites co-immunoprecipitated with Cdc5-V5 in wild-type and *spo13Δ* strains. (E) Abundance of Clb1 phospho-sites co-immunoprecipitated with anti-V5 in no tag, Cdc5-V5 ("wild type"), and *spo13Δ* Cdc5-V5 strains. Values are log2 transformed and mean-centred by phospho-site, calculated by subtracting the mean abundance across all samples from the abundance in a given sample. Error bars represent the 95% confidence interval around the mean. Data information for (A–E): Data from n = 3 biological replicates. Statistics: DEP R package function test_diff, which tests for differential expression by empirical Bayes moderation of a linear model on the predefined contrasts. Source data are available online for this figure.

## Spo13 promotes phosphorylation of [DEN]x[ST] *[FG] motif sites

In contrast to *spo13Δ*, depletion of Cdc5$^{Polo}$ leads to metaphase I arrest (Lee and Amon, 2003; Clyne et al, 2003), indicating that Spo13 must control phosphorylation of only a subset of Cdc5$^{Polo}$ target sites. Indeed, only ~10–12% of Cdc5$^{Polo}$-motif-matching sites had decreased phosphorylation in *spo13Δ* in both the metaphase I arrest and the timecourse experiments (Fig. EV5A,B). This raises the question of how Spo13 enhances Cdc5$^{Polo}$ kinase phosphorylation for only a subset of substrates. We hypothesised that Spo13 may direct Cdc5$^{Polo}$ phosphorylation to a subset of targets containing a preferred motif. To test this idea, we searched for sub-motifs among the differentially phosphorylated Cdc5$^{Polo}$ motif-matching sites, matching the pattern [DEN]x[ST]*, in wild-type versus *spo13Δ* or *spo13-m2* in metaphase I and discovered a preference for phenylalanine (F) or glycine (G) in the +1 position among sites with reduced phosphorylation in either *spo13Δ* or *spo13-m2* (Fig. 7E,F). This suggests that Spo13, either directly or indirectly, promotes Cdc5$^{Polo}$-directed phosphorylation of substrates carrying the motif [DEN]x[ST]*[FG].

## [DEN]x[ST]*F motif site phosphorylation depends on Cdc5$^{Polo}$ activity in vivo

To more directly test whether Cdc5$^{Polo}$ is responsible for [DEN]x[ST]*[FG] motif site phosphorylation, we analysed the phospho-proteome upon acute inhibition of Cdc5$^{Polo}$ by using the analogue-sensitive allele *cdc5-as* (Paulson et al, 2007). We released wild-type or *cdc5-as* cells from prophase arrest, added inhibitor after 30 min and then harvested cells 30 min later in prometaphase I (Appendix Fig. S8C). We quantified 4117 proteins and 5848 phosphorylation sites in 2 replicates each of wild-type and *cdc5-as* cells (Appendix Fig. S9B). We found that 706 phospho-sites decreased (~12%) while 1130 (~19%) increased more than 1.5-fold in *cdc5-as*, with the remainder (4012; ~69%) having no significant difference (Fig. EV6A; Dataset EV14). Asparagine (N), glutamic acid (E) or aspartic acid (D) were enriched at the −2 position surrounding phospho-sites exhibiting decreased phosphorylation in *cdc5-as* (Fig. EV6B), as expected from previous analysis of Polo specificity in mitotic cells (Santamaria et al, 2011). Sites with increased phosphorylation in *cdc5-as* showed enrichment for serine at −4 through −2 (Fig. EV6C) and the dominant kinase responsible is unclear.

A slight preference for Polo kinase to phosphorylate substrates with hydrophobic or aromatic residues at +1 has been previously noted (Mok et al, 2010; Santamaria et al, 2011). Consistently, we

observed an enrichment for hydrophobic and aromatic residues at +1 among Cdc5$^{Polo}$-dependent sites (Fig. EV6D). Notably, we observed that a significant number of [DEN]x[ST]*F, but not [DEN]x[ST]*G, motif sites depend on Cdc5$^{Polo}$ activity for their phosphorylation (Fig. EV6D). Given that Cdc5-as was acutely inhibited in our experiment, Cdc5$^{Polo}$ may directly phosphorylate [DEN]x[ST]*F sites. In contrast, the [DEN]x[ST]*G motif, the phosphorylation of which depends on Spo13 in metaphase I (Fig. 7E,F) may not be a direct Cdc5$^{Polo}$ substrate or its phosphorylation may be more stable.

Arginine (R) was enriched at the −3 position among sites found to be decreased in *cdc5-as* (Fig. EV6D), perhaps reflecting cross-talk between Cdc5$^{Polo}$ signalling and basophilic kinases such as Ipl1$^{AuroraB}$, while +1 proline was significantly under-represented (Fig. EV6D), in agreement with the finding that Polo tolerates most amino acids at that position except for proline (Santamaria et al, 2011). In conclusion, Cdc5$^{Polo}$ is responsible for the phosphorylation of a significant number of [DEN]x[ST]* as well as [DEN]x[ST]*F motif sites but does not promote [ST]*P motif phosphorylation.

## Substrates whose phosphorylation depends on Spo13 are enriched for the polo-box binding motif S[ST]P

Cdc5$^{Polo}$ interacts with substrates through its polo-box domain (PBD), consisting of two polo-boxes that recognise phospho-serine or phospho-threonine preceded by serine (S[ST]*) in substrates which have undergone a priming phosphorylation by another kinase such as Cdk (Elia et al, 2003). As mentioned earlier, Spo13 possesses a S[ST]*P motif (132-STSTP-136) through which Cdc5$^{Polo}$ binding occurs and the *spo13-m2* mutant (*spo13-S132T,S134T*) disrupts this motif (Matos et al, 2008). This suggests that, in the Spo13-Cdc5$^{Polo}$ complex, substrate recognition through the PBD is blocked by Spo13 binding. This raises the question of whether substrates whose phosphorylation is promoted by Spo13 rely on the canonical mode of Cdc5$^{Polo}$ binding via the PBD. Interestingly, a greater-than-expected fraction of proteins with decreased phosphorylation in *spo13Δ* and *spo13-m2* cells contained the S[ST]P motif, regardless of whether it was phosphorylated (Appendix Fig. S12A,B). Though not significant, the subset of proteins with Cdc5$^{Polo}$ motif phosphorylation ([DEN]x[ST]*) were also more likely to contain a S[ST]P motif in *spo13Δ* but not *spo13-m2* (Appendix Fig. S12A,B). Phosphorylated S[ST]*P sites were also enriched among phospho-sites with decreased phosphorylation in *spo13Δ* and *spo13-m2* cells (Appendix Fig. S12C,D) and similar trends were observed when the same analysis was performed on the timecourse data (Appendix Fig. S12E,F). Thus, S[ST]*P motifs are frequently present in likely Cdc5$^{Polo}$ substrates whose

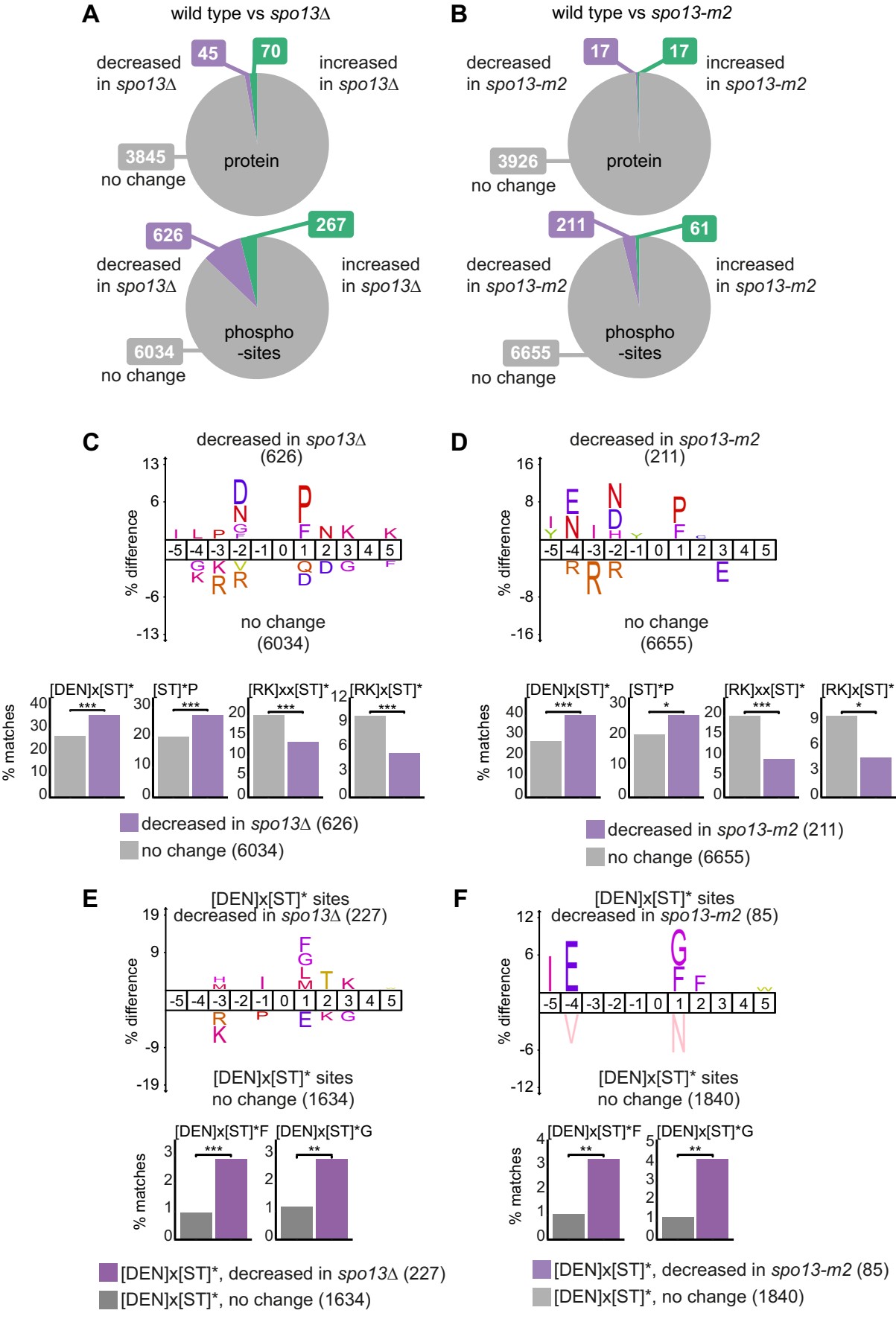

**Figure 7.   At metaphase I, Cdc5$^{Polo}$ kinase phosphorylation is decreased in *spo13Δ* and *spo13-m2* cells.**

(A) Proportion of proteins and phospho-sites which significantly vary between wild-type and *spo13Δ* in metaphase I-arrested cells. See also Datasets EV10 and EV12. (B) Proportion of proteins and phospho-sites which significantly vary between wild-type and *spo13-m2* in metaphase I-arrested cells. See also Datasets EV11 and EV13. (C) IceLogo of motifs enriched surrounding phospho-sites decreased in *spo13Δ* in metaphase I-arrested cells. Bar charts below show percent motif matches in the indicated groups of sites. Data information: Statistics: Fisher's exact test. ***$P < 0.001$. (D) IceLogo of motifs enriched surrounding phospho-sites decreased in *spo13-m2* in metaphase I-arrested cells. Bar charts below show percent motif matches in the indicated groups of sites. Data information: Statistics: Fisher's exact test. ***$P < 0.001$, *$P < 0.05$. (E) IceLogo of motifs enriched surrounding [DEN]x[ST]* phospho-sites decreased in *spo13Δ* in metaphase I-arrested cells. Bar charts below show percent motif matches in the indicated groups of sites. Data information: Statistics: Fisher's exact test. ***$P < 0.001$, **$P < 0.01$. (F) IceLogo of motifs enriched surrounding [DEN]x[ST]* phospho-sites decreased in *spo13-m2* in metaphase I-arrested cells. Bar charts below show percent motif matches in the indicated groups of sites. Data information: Statistics: Fisher's exact test. **$P < 0.01$. Source data are available online for this figure.

phosphorylation is promoted by Spo13. Therefore at least in principle, phosphorylation could occur through canonical substrate binding via the PDB, though how Spo13 enables this remains unresolved.

## Spo13-dependent phosphorylation occurs on chromosome-related proteins

To understand whether phospho-proteins regulated by Spo13 at metaphase I have shared functions, we performed GO term enrichment analysis. Among proteins with decreased phosphorylation in *spo13Δ* or *spo13-m2* vs wild type in metaphase I arrest, the terms "chromatin remodelling" and "RNA biosynthetic process" were enriched (Fig. EV5C). For proteins with increased phosphorylation in *spo13Δ*, the cytoskeleton-related terms "prospore membrane" and "cytoskeleton organisation" were most enriched (Fig. EV5C). To look more specifically at proteins regulated by Cdc5$^{Polo}$ phosphorylation, we restricted the GO term analysis to proteins with phosphorylation on sites matching the canonical Cdc5$^{Polo}$ motif, [DEN]x[ST]*. Reflecting the primary effect of Spo13 on Cdc5$^{Polo}$, nearly the same terms were enriched (Fig. 8A). Further restricting our analysis to the 74 proteins phosphorylated at metaphase I on the [DEN]x[ST]*F motif promoted by Spo13-Cdc5$^{Polo}$ identified above, GO terms enriched included "meiotic chromosome separation", "chromosome organisation" and "chromatin binding" (Fig. 8B). The majority (57%) of these proteins phosphorylated on [DEN]x[ST]*F in the metaphase I arrest were also phosphorylated in our timecourse analysis (42/74) (Fig. 8C), consistent with Spo13 exerting its effects in metaphase I and providing confidence that these sites represent functionally important substrates.

## Spo13 promotes phosphorylation of [DEN]x[ST]*F sites in meiosis I

Cdc5$^{Polo}$ is induced at prophase I exit and Spo13 is degraded in anaphase I, therefore Spo13 and Cdc5$^{Polo}$ co-exist only during prometaphase I and metaphase I. Accordingly, if [DEN]x[ST]*F phosphorylation is promoted by Spo13-directed Cdc5$^{Polo}$ activity, it is expected to be maximal during prometaphase/metaphase I. We noticed in the wild-type phosphorylation timecourse data that among the three clusters enriched for the canonical Cdc5$^{Polo}$ motif [DEN]x[ST]* (clusters 3, 4, and 5, Figs. 3D, EV2, and 8D), only the cluster that is specific to metaphase I is significantly enriched for the variant [DEN]x[ST]*F motif (cluster 3, Figs. 3D, EV2, and 8E). To investigate the timing of phosphorylation of these sites in more detail, we plotted the average abundance of all sites matching the [DEN]x[ST]*F and/or

[DEN]x[ST]*G motifs, as well as the extended [DEN]x[ST]*[LIMYF] motif identified in previous studies, across the timecourse in wild-type and *spo13Δ* (Figs. 8F and EV5D) (Mok et al, 2010; Santamaria et al, 2011). In wild type, all motifs showed two peaks of abundance in metaphase I-anaphase I (75–90 min) and in metaphase II (135 min) (Figs. 8F and EV5D). With the exception of [DEN]x[ST]*G (Fig. 8F), these peaks were more pronounced for all extended motifs, compared to the canonical motif. Interestingly, phosphorylation of [DEN]x[ST]*F was increased in meiosis I relative to meiosis II (Fig. 8F), indicating that F in the +1 position is the best predictor of meiosis I-specific phosphorylation of the Cdc5$^{Polo}$ consensus. Consistently, in *spo13Δ* the pronounced meiosis I peak of phosphorylation was lost on sites with F in the +1 position (Fig. 8F). Overall, our data indicate that Cdc5$^{Polo}$ preferentially phosphorylates sites with the motif [DEN]x[ST]*F in meiosis I and that this is enhanced by Spo13.

## Discussion

Here we describe the time-resolved global proteome and phospho-proteome of the meiotic divisions, providing unprecedented insight into the protein and phosphorylation changes that govern this specialised cell cycle. Our datasets provide a valuable resource to discover the mechanisms underlying the regulation of key meiotic processes. While protein changes were limited to a small number of key regulators, we found that distinct groups of phosphorylations characterised each stage of the meiotic divisions. Matching phosphorylations to consensus motifs, the identity of which is supported by our kinase inhibition experiments, revealed that Cdc28$^{Cdk1}$ and Cdc5$^{Polo}$ each direct two waves of phosphorylation, corresponding to meiosis I and II. Our data also suggest that while phosphorylation by Cdc28$^{Cdk1}$ declines in anaphase I, Cdc5$^{Polo}$ is active in anaphase I, and that a distinct set of proteins are phosphorylated in meiosis I and meiosis II. We found that Spo13 is required for a subset of Cdc5$^{Polo}$-directed phosphorylation in meiosis I and for two waves of Cdc28$^{Cdk1}$ phosphorylation. Analysis of metaphase I-arrested cells identified a specific motif, [DEN]x[ST]*F, which is enriched in the wild-type phosphoproteome at this time but strongly depleted from *spo13Δ* and *spo13-m2* cells. Therefore, by facilitating Cdc5$^{Polo}$-directed phosphorylation at metaphase I, particularly of substrates containing the [DEN]x[ST]*F motif, Spo13 establishes the meiotic programme.

### Two waves of Cdk phosphorylation in meiosis

A longstanding question is how cells progress through the meiosis I to meiosis II transition without resetting replication origins. The

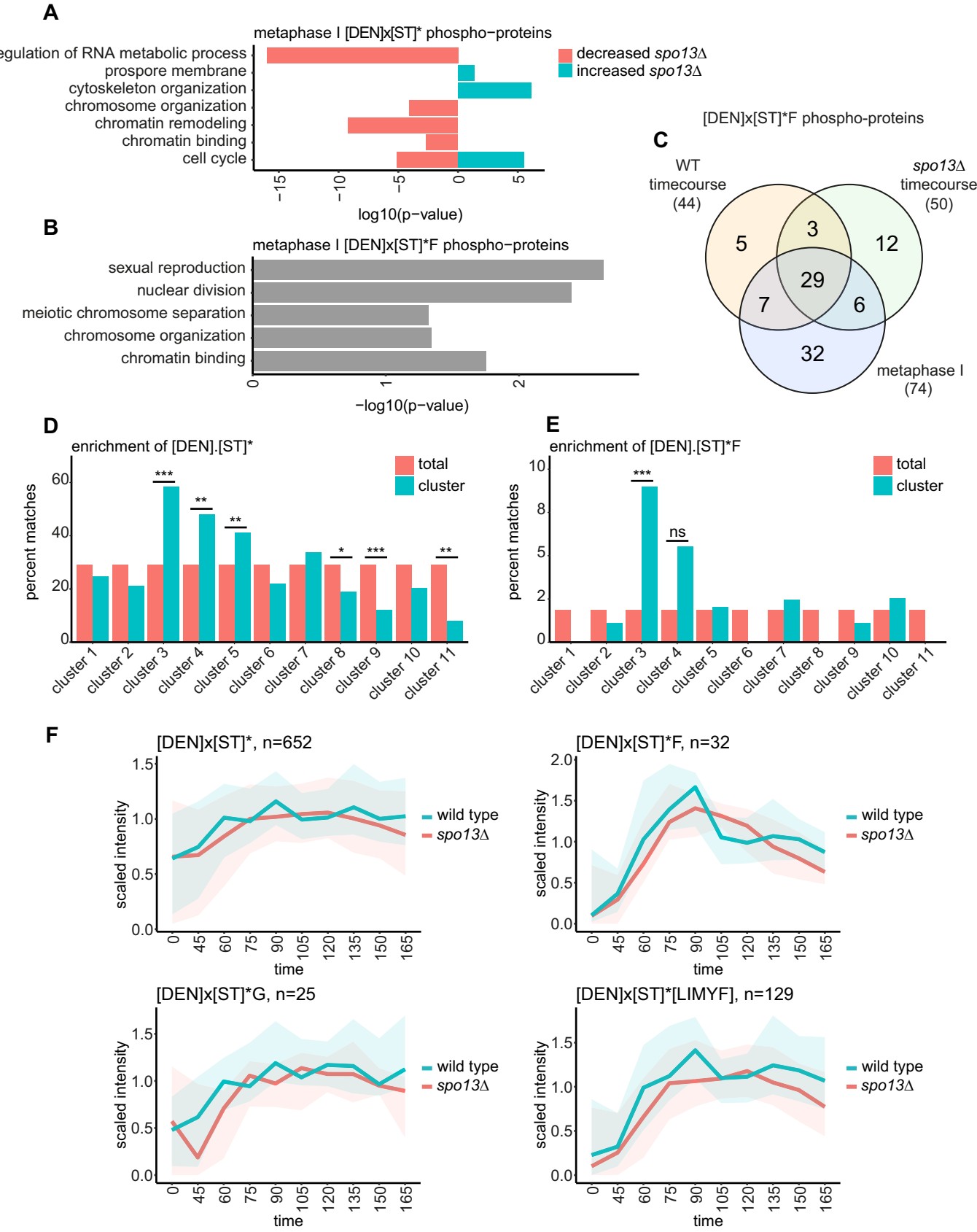

Figure 8.   Spo13 promotes [DEN]x[ST]*F phosphorylation in metaphase I.

(A) Bar graph of selected GO terms enriched among proteins with phosphorylated [DEN]x[ST]* sites with significantly different abundance between wild type and spo13Δ in the metaphase I arrest dataset. Data information: Statistics: Cumulative hypergeometric test followed by correction for multiple testing (gprofiler2 R package gost function default settings). (B) Bar graph of selected GO terms enriched among proteins with phosphorylated [DEN]x[ST]*F sites in the metaphase I arrest dataset. Data information: Statistics: Cumulative hypergeometric test followed by correction for multiple testing (gprofiler2 R package gost function default settings). (C) Proportion of phosphoproteins with phosphorylated [DEN]x[ST]*F motifs detected in the wild-type timecourse, spo13Δ timecourse, and both wild type and spo13Δ in the metaphase I arrest experiment. (D) Percent [DEN]x[ST]* motif matches in the indicated groups of sites. Clusters refer to wild-type phospho-site clustering in Figs. 3 and EV2. Data information: Statistics: Fisher's exact test. ***P < 0.001, **P < 0.01, *P < 0.05. (E) Percent [DEN]x[ST]*F motif matches in the indicated groups of sites. Clusters refer to wild-type phospho-site clustering in Figs. 3 and EV2. Data information: Statistics: Fisher's exact test. ***P < 0.001. (F) Abundance of phospho-sites matching the indicated motifs among sites detected in both replicates of wild-type and spo13Δ across the timecourse. Source data are available online for this figure.

prevailing hypothesis is that Cdks retain activity towards certain substrates at this transition, however, this has been challenging to address because of poor meiotic synchrony coupled with the short time interval between anaphase I and metaphase II. Nevertheless, accumulating evidence indicates that Cdks are at least partially inactivated between meiosis I and II (Furuno et al, 1994; Iwabuchi et al, 2000; Swartz et al, 2021; Carlile and Amon, 2008) and changes in the abundance and activity of the B-type cyclins present during the meiotic divisions certainly play a role, however the underlying mechanisms are not well understood (Carlile and Amon 2008). We showed that phosphorylation of the cyclin Clb1, which promotes its stability, $Cdc28^{Cdk1}$ activity and APC function (Rojas et al, 2023), depends on Spo13, but the implications for this in control of the meiosis I to II transition require further investigation.

A second hypothesis for how the two-division programme is established is that the activity of a distinct kinase could maintain some phosphorylation at the meiosis I to II transition. In support of this idea, Ime2 kinase is at least partially active as cells transition into meiosis II (Berchowitz et al, 2013) (see also Fig. EV2) and the sites it phosphorylates are resistant to dephosphorylation by the Cdc14 phosphatase (Holt et al, 2007). However, to date only a handful of in vivo Ime2 substrates have been identified (Berchowitz et al, 2013) and its consensus motif partially overlaps with that of many basophilic kinases (Holt et al, 2007; Mok et al, 2010). Consequently, very few bone fide Ime2-specific motif, RPx[ST]*, phospho-sites were found in our dataset (Appendix Fig. S3B), precluding further conclusions on the extent of Ime2 activity at the global level. However, a recent study demonstrated that $Cdc28^{Cdk1}$and Ime2 cooperatively inhibit helicase loading at origins between the meiotic divisions (Phizicky et al, 2018), providing evidence for this model.

A third, not mutually exclusive, hypothesis for how cells progress into meiosis II without an intervening S phase posits that selective dephosphorylation of substrates could temper a loss of kinase activity at the meiosis I to II transition. Previous studies have suggested that Cdc14 phosphatase may have a higher affinity for strict versus minimal $Cdc28^{Cdk1}$ consensus sites (Bremmer et al, 2012) and differential dephosphorylation kinetics of these motifs has been observed in a phosphoproteomic analysis of mitotic exit, and attributed to phosphatase specificity (Touati et al, 2018). In support of this idea, we discovered that phosphorylation of strict $Cdc28^{Cdk1}$ consensus sites falls substantially at meiosis I exit, while the minimal $Cdc28^{Cdk1}$ consensus phospho-sites do not. This could indicate that phosphatases active at the meiosis I to II transition have a preference for $Cdc28^{Cdk1}$ sites with lysine or arginine in the +3 position, while sites with a different residue at +3 are protected from dephosphorylation. However, our phosphoproteomic analysis

of cdc28-as inhibition found that [ST]*Px[KR] inhibition is the best predictor of $Cdc28^{Cdk1}$ activity. Therefore, an alternative, more straightforward explanation is that the majority of [ST]*P sites are phosphorylated by other kinases which remain active at this transition. Directed mechanistic studies are required to test these ideas and also to understand how Cdks are permitted to rise again for entry into meiosis II.

## Spo13 promotes $Cdc5^{Polo}$-dependent phosphorylation of [DEN]x[ST]*F sites in meiosis I

One hypothesis inspired by our data is that Spo13 directs $Cdc5^{Polo}$ preferentially to substrates carrying a variant of the $Cdc5^{Polo}$ consensus motif with phenylalanine in the +1 position, [DEN]x[ST]*F. This proposal is supported by four lines of evidence. First, [DEN]x[ST]*F was strongly depleted from the spo13Δ and spo13-m2 metaphase I phosphoproteome. Second, acute kinase inhibition demonstrated that $Cdc5^{Polo}$ is responsible for [DEN]x[ST]*F motif phosphorylation. Third, in the wild-type timecourse experiment, among the clusters where the $Cdc5^{Polo}$ consensus was enriched, only those with high abundance in metaphase I showed enrichment for the F at +1. Fourth, abundance of [DEN]x[ST]*F reached a clear peak at metaphase I in wild type, but not in spo13Δ. The fact that Spo13 is degraded in anaphase I and does not reaccumulate in meiosis II (Sullivan and Morgan, 2007; Katis et al, 2004), together with our finding that [DEN]x[ST]*F phosphorylation peaks at metaphase I in wild-type provide further support that Spo13 and $Cdc5^{Polo}$ are together responsible for this phosphorylation.

Might Spo13 influence the substrate choice of $Cdc5^{Polo}$ and, if so, how? The question is particularly intriguing since Spo13 contains an S[ST]P motif which is required for binding to $Cdc5^{Polo}$ (Matos et al, 2008). This suggests that $Cdc5^{Polo}$ uses its Polo Binding Domain (PBD) to bind to Spo13 in the same way it normally binds to substrates (Elia et al, 2003). Our phosphoproteome of spo13-m2 mutant metaphase I cells, in which this motif is mutated, showed similar, albeit more moderate, trends as in spo13Δ, suggesting that Spo13 predominantly influences [DEN]x[ST]* phosphorylation through its binding to the $Cdc5^{Polo}$ PBD. In apparent contradiction, however, we found that the proteins that depend on Spo13 for their phosphorylation frequently harboured a polo-box binding motif S[ST]P. Therefore, either both Spo13 and substrate bind through S[ST]*P-PBD interactions but not simultaneously, or only Spo13 binds in this way, in which case $Cdc5^{Polo}$ must recognise its substrate through a S[ST]P-independent mechanism. Indeed, there is precedent for $Cdc5^{Polo}$ using a different surface of its PBD to associate with binding partners, an interaction that can occur coincidentally with binding to a substrate in the canonical manner

(Chen and Weinreich, 2010; Almawi et al, 2020). Therefore, it remains possible that Spo13-Cdc5$^{Polo}$ can stay associated while Cdc5$^{Polo}$ interacts with another substrate. However, the question remains how Spo13 binding would influence Cdc5$^{Polo}$ to promote the preferred phosphorylation of [DEN]x[ST]*F sites. One possibility is that Cdc5$^{Polo}$ intrinsically prefers to phosphorylate [DEN]x[ST]*F motifs and relies on docking through S[ST]P-PBD interactions to phosphorylate less favourable motifs. In this case, Spo13 would block Cdc5$^{Polo}$ binding to substrates through S[ST]P sites, thereby imposing a requirement on the strict consensus [DEN]x[ST]*F for phosphorylation to occur. Structural and biochemical studies are required to address these important questions.

## The phosphorylation events that programme homologue segregation

What are the functional consequences of Spo13 rewiring the phosphoproteome? We find that chromosome and cell cycle-associated proteins are enriched within the Spo13-Cdc5$^{Polo}$ phosphoproteome. This is consistent with both the documented roles of Spo13 in sister kinetochore monoorientation, cohesion protection and the execution of two meiotic divisions (Lee et al, 2004; Katis et al, 2004) and the fact that at least a fraction of Spo13 is localised to chromosomes (Galander et al, 2019). Our study therefore provides a starting point for future investigations to delineate the molecular mechanisms underlying the functional consequences of Spo13-Cdc5$^{Polo}$-dependent phosphorylation. We note, however, that Cdc5$^{Polo}$ also has meiotic functions independent of Spo13, implying that only a fraction of cellular Cdc5$^{Polo}$ is associated with Spo13 and raising the possibility that Spo13 is rate-limiting. The observation that the over-expression of Spo13 blocks cells in mitotic metaphase (Shonn et al, 2002; Lee et al, 2002; McCarroll and Esposito, 1994; Maier et al, 2021) could suggest titration of Cdc5$^{Polo}$ away from key targets, in support of this idea. Understanding the balance between free and Spo13-bound Cdc5$^{Polo}$ will be an important avenue of future investigation.

Although meiotic errors are frequent, causing fertility issues and developmental disorders in humans, the molecular basis of meiosis is much less understood than mitosis, in part due to the scarcity of meiotic material. Our proteomics and phosphoproteomics study in budding yeast has provided a rich dataset that can help to bridge this gap to discover key molecular mechanisms that could be relevant for human fertility.

## Methods

### Yeast strains

Yeast strains used in this study were derivatives of SK1 and are listed in Dataset EV15. *pCLB2-CDC20* (Lee and Amon, 2003), *pGAL1-NDT80, pGPD1-GAL4.ER* (Benjamin et al, 2003), *spo13-m2* (Matos et al, 2008), *cdc28-as* (Cdc28-F88G) (Bishop et al, 2000) and *cdc5-as* (Cdc5-L158G) (Paulson et al, 2007) were described previously. For the Cdc5 IP, Cdc5 was tagged by standard PCR methods. Yeast strains are available from the corresponding author without restriction.

### Spindle immunofluorescence

Meiotic spindles were visualised by indirect immunofluorescence as described in Barton et al, 2022. A rat anti-tubulin primary antibody (AbD serotec) at 1:50 dilution and an anti-rat FITC conjugated secondary antibody (Jackson Immunoresearch) at 1:100 dilution were used. In total, 200 cells were counted at each timepoint and/or for each sample. A Zeiss Axioplan Imager Z2 fluorescence microscope with a 100x Plan ApoChromat NA 1.45 oil lens with a Teledyne Photometrics Evolve 512 EMCCD camera was used to take the representative images.

### Meiotic prophase block-release timecourse

Cells were induced to undergo meiosis as described by Barton et al, 2022 and *pGAL-NDT80* prophase block-release experiments were performed as outlined in Carlile and Amon, 2008. Briefly, strains were patched from −80 °C stocks to YPG agar (1% Bacto yeast extract, 2% Bacto peptone, 2.5% glycerol, 0.3 mM adenine, 2% agar) plates. After ~16 h, cells were transferred to 4% YPDA agar (1% Bacto yeast extract, 2% Bacto peptone, 4% glucose, 0.3 mM adenine, 2% agar) at 30 °C. After 8–16 h, cells were inoculated into YPDA media and grown for 24 h at 30 °C with shaking at 250 rpm. Next, BYTA (1% Bacto yeast extract, 2% Bacto tryptone, 1% potassium acetate, 50 mM potassium phthalate) cultures were prepared to $OD_{600} = 0.3$ and grown at 30 °C with shaking at 250 rpm overnight (~16 h). The next morning, cells were washed twice in sterile water and resuspended in sporulation medium at $OD_{600} = 2.0$. After 6 h in sporulation media, 1 μM β-estradiol was added to release cells from prophase and samples were collected for immunofluoresence every 15 min until 180 min, and then every 30 min for a further hour. For the preparation of protein extracts for mass spectrometry, 10 ml samples were collected at time 0 and at 45–165 min after prophase release by centrifugation at 3000 rpm for 3 min and the supernatants removed. Cells were resuspended in 5% TCA and incubated on ice for ~10 min. Next the samples were centrifuged again for 3 min at 3000 rpm and the supernatant removed. The samples were transferred to 2 ml Fastprep tubes (MP biomedicals) and spun in a microcentrifuge at 13.2k rpm for 1 min. The supernatant was removed by aspiration and the cell pellets drop frozen in liquid nitrogen before being stored at −80 °C before further processing.

### Analogue-sensitive kinase experiments

For the *cdc28-as* experiment, *pGAL-NDT80* strains were induced to undergo meiosis as described by Barton et al, 2022 and cultured in sporulation medium for 6 h to arrest in prophase. For the "Meiosis I" samples, 1 μM β-estradiol and DMSO or 5 μM 1NM-PP1 was added to wild-type or *cdc28-as* strains respectively and samples were collected after 75 min, when wild-type cells would be at the peak of metaphase I by spindle IF. For the "Meiosis II" samples, 1 μM β-estradiol was added after 6 h in sporulation medium to release from prophase and after 90 min for wild type or 105 min for *cdc28-as* when the cells were in anaphase I by spindle immuno-fluorescence, DMSO or 5 μM 1NM-PP1 was added to wild-type or *cdc28-as* strains respectively and samples were collected 15–30 min later, depending on when the majority of cells of the same strain had metaphase II spindles in additional experiments where DMSO/1NM-PP1 was not added.

For the *cdc5-as* experiment, *pGAL-NDT80* strains were induced to undergo meiosis as described by Barton et al, 2022 and cultured in sporulation medium for 6 h to arrest in prophase. In total, 1 μM β-estradiol was added to release cells from prophase, 5 μM CMK as added 30 min after and samples were collected at 60 min from prophase release, corresponding to prometaphase by spindle IF.

## Immunoprecipitation of Cdc5 for label-free quantification mass spectrometry (LFQMS)

*pGAL-NDT80* strains were induced to undergo meiosis as described by Barton et al, 2022 and cultured in sporulation medium for 6 h to arrest in prophase. In total, 1 μM β-estradiol was added to release cells from prophase and samples from the wild-type no tag strain were collected at 90 min and the Cdc5-V5 or *spo13Δ* Cdc5-V5 samples were collected at 105 min. Cell stage at metaphase I was verified by spindle IF.

Cryo-lysis was performed with a freezer mill (Spex 6875) with eight rounds of 2 min at ten cycles/second, with 2 min rests between grinding. α-V5 antibodies were conjugated to Protein G Dynabeads (ThermoFisher) by crosslinking with dimethyl pimeli-midate (DMP, ThermoFisher). Immunoprecipitation was performed on all of the cell lysate prepared from 800 mL meiotic culture at $OD_{600} = 2.0$. Specifically, 100 μl of V5-conjugated Dynabeads were incubated with approximately 5 mL cell extract containing ~80 mg of protein for 2 h at 4 °C with rotation. This was followed by two washes in buffer H 0.15 M (25 mM HEPES (pH 8.0), 2 mM $MgCl_2$, 0.1 mM EDTA (pH 8.0), 0.5 mM EGTA-KOH (pH 8.0), 15% glycerol, 0.1% NP-40, 150 mM KCl) containing 2 mM DTT and supplemented with phosphatase inhibitors (2 mM β-glycerophosphate, 1 mM $Na_4P_2O_7$, 5 mM NaF, 0.1 mM $Na_3VO_4$), protease inhibitors (2 mM final AEBSF, 0.2 μM microcystin and 10 μg/mL each of "CLAAPE" protease inhibitors (chymostatin, leupeptin, antipain, pepstatin, E-64)), followed by one wash in buffer H 0.15 M alone. Proteins were eluted in two rounds with 0.1% RapiGest (Waters) in 50 mM Tris pH 8.0 by incubating at 50 °C for 10 min with mixing and then drop frozen in $LN_2$ and stored at −80 °C.

The eluate samples were prepared for MS by a filter-aided sample preparation (FASP) method, as described with minor modifications (Wiśniewski et al, 2009). Samples were reduced with 25 mM DTT at 80 °C for 1 min, then denatured by addition of urea to 8 M. Sample was applied to a Vivacon 30k MWCO spin filter (Sartorius) and centrifuged at $12.5k \times g$ for 15–20 min. Protein retained on the column was then alkylated with 100 μL of 50 μM iodoacetamide (IAA) in buffer A (8 M urea, 100 mM Tris pH 8.0) in the dark at room temperature for 20 min. The column was then centrifuged as before, and washed with 100 μL buffer A, then with $2 \times 100$ μL volumes of 50 mM ammonium bicarbonate (AmBic). In total, 3 μg/μL trypsin (Pierce) in 0.5 mM AmBic was applied to the column, which was capped and incubated at 37 °C overnight. Digested peptides were then spun through the filter, acidified with trifluoroacetic acid (TFA) to pH <= 2, loaded onto manually-prepared and equilibrated C18 stage tips (Rappsilber et al, 2003) washed with 100 μL 0.1% TFA, and stored at −20 °C prior to MS analysis.

## Metaphase I arrest experiment

Cells were induced to undergo meiosis as described by Barton et al, 2022 and cultured in sporulation medium for 6.5 h before harvest.

10 ml samples were collected by centrifugation and resuspended in 5% TCA on ice for 10 min. Next the samples were centrifuged again for 3 min at 3000 rpm and the supernatant removed. The samples were transferred to 2 ml Fastprep tubes (MP biomedicals) and spun in a microcentrifuge at 13.2k rpm for 1 min. The supernatant was removed and the cell pellets drop frozen in liquid nitrogen before being stored at −80 °C before further processing.

## Sample preparation for Tandem Mass Tag (TMT) mass spectrometry

Samples from all experiments except the Cdc5 IP were prepared and analysed using the following tandem mass tag (TMT) mass spectrometry methods. Cell pellets stored at −80 °C were placed on ice and then 500 μl ice-cold acetone was added and the samples were vortexed for a few seconds and placed at −20 °C for ~1 h while urea lysis buffer (8 M urea, 75 mM NaCl, 50 mM HEPES pH 8.0) was prepared. Samples were spun in a chilled 4 °C microcentrifuge at 13.2k rpm for 10 min before the supernatants were removed and the pellets were dried in the hood for 10 min. Each sample was then resuspended in 200 μl freshly prepared 8 M urea lysis buffer, supplemented with 2 mM beta glycerophosphate, 1 mM Na pyrophosphate, 5 mM NaF, 2 mM AEBSF, 2× CLAAPE (10 μg/ul each of Chymostatin, Leupeptin, Antipain, Aprotinin, Pepstatin A, E-64 protease inhibitor), 0.8 mM $NaVO_4$, 0.2 μM microcystin-LR, and 1× Roche protease inhibitor cocktail, by pipetting up and down. Silica beads were added and cells were lysed by bead-beating in a FastPrep (MP Biomedicals) machine at 4 °C, with $4 \times 45$ s rounds of lysis and 2 min on ice in between each round. Cell lysates were separated from beads by poking a hole in the bottom of each tube with a red-hot needle and placing the tube on top of a new Fastprep tube and briefly spinning them for ~20 s in a centrifuge. Next, the samples were spun for 15 min at 7k rpm in a chilled 4 °C microcentrifuge, and the clarified lysate super-natants were transferred to protein lo-bind Eppendorf tubes. A BCA assay (Pierce) was carried out, following the manufacturer's instructions, to determine protein concentration.

A TMT10plex method was used for all TMT experiments except for the *cdc28-as* experiment which was done as a part of a TMT16plex. For the TMT10plex experiments, 400 μg of each protein sample was processed and for the TMT16plex 250 μg of each protein sample was processed in the same way. Samples were reduced by adding 5 mM DTT and incubating at 37 °C for 15 min with gentle shaking at 500 rpm in an Eppendorf shaker/incubator. Next, samples were alkylated in 10 mM iodoacetamide (IAA) for 30 min in the dark. Then, 15 mM DTT was added and samples were incubated for a further 15 min in the dark. Finally, 5x volume of ice-cold acetone was added to each sample, briefly vortexed, and stored at −20 °C overnight (~16 h).

The next day, samples were spun in a chilled 4 °C microcentrifuge at $8000 \times g$ for 10 min, the supernatant removed and the pellets dried in the hood for 10 min. TMT10plex samples were resuspended in 90 μl of 100 mM TEAB (Thermo Scientific), and TMT16plex samples were resuspended in 100 μl 100 mM TEAB. Samples were digested with 1:40 Trypsin (w/w) in two rounds to facilitate digestion, with half of the Trypsin added and samples incubated 37 °C for 4 h, then the remaining half Trypsin added and samples mixed and incubated 37 °C overnight (16 h). The next morning, samples were clarified by spinning in a microcentrifuge at

4k rpm for 10 min at room temperature and the supernatant removed to fresh tubes. The final volumes were brought to ~100 μl in 100 mM TEAB. For TMT10plex, each 0.8 mg TMT reagent (Thermo Scientific) was resuspended in 41 μl acetonitrile, mixed with the 0.4 mg peptide sample (2:1 TMT:peptide w/w). For TMT16plex, each 0.5 mg TMT reagent (Thermo Scientific) was resuspended in 20 μl acetonitrile and mixed with the 0.25 mg peptide sample (2:1 TMT:peptide w/w). TMT label and peptides were gently mixed by shaking at 400 rpm at 25 °C for 1 h. Reactions were quenched with 50 mM Tris pH 8.0 and shaken at 400 rpm, 25 °C for 15 min. Following TMT labelling, all samples were combined together in one tube and acidified to pH <3 with formic acid, dried in a vacuum centrifuge and stored at −80 °C. The sample was resuspended in 0.1% formic acid, pH <3 and then desalted using a 500 mg C18 SepPak cartridge (Waters). To remove free TMT from samples, an extra wash with 0.5% acetic acid was added to the protocol and elution was done in 70% acetonitrile, diluted in 0.2% formic acid.

For the TMT16plex experiment, the peptide sample was eluted from the C18 cartridge in phospho-enrichment load buffer (80% ACN, 5% TCA and 5% glycolic acid) and phospho-enrichment carried out directly. ~100 μg peptide sample was reserved for the non-phospho ("N") sample and the remaining peptides were incubated with 1 mg of MagReSyn Ti-IMAC beads for 20 min at 25 °C. Beads were washed for 2 min with 80% acetonitrile + 1% TFA. This was followed by two 2-min washes in 10% acetonitrile + 0.2% TFA. Phosphorylated peptides were eluted for 15 min twice in 1% ammonium hydroxide. This was followed by a second elution for 1 h in 1% ammonium hydroxide 50% acetonitrile. The sample was dried and desalted again on a 100 mg C18 SepPak cartridge (Waters) before fractionation.

For the TMT10plex experiment, the phospho-enrichment of the samples was carried out after the following fractionation method. Samples were resuspended in 10 mM ammonium formate pH 9.0 and fractionated by high pH reversed-phase (HPRP) chromatography using an Ultimate 3000 HPLC (ThermoFisher Scientific). Peptides were loaded onto a XBridge Peptide BEH C18 130 Å 3.5 μm 4.6 × 150 mm column (Waters) and eluted at 1 ml/min using a constant 10 mM ammonium formate pH 9.0 and a multistep gradient from 2% to 50% acetonitrile over 10 min, 50% to 80% gradient for a further 1 min, followed by an 80% column wash and re-equilibration. 72 fractions were collected at 8.5 s intervals and concatenated into 12 fractions and dried.

For the TMT16plex experiment, for both the "N" and "PE" peptide sample, the same HPLC and buffer conditions were used with the following modifications. Peptides were loaded onto a Hypersil GOLD C18 175 Å 1.9 μm 1 × 100 mm column (Thermo-Fisher Scientific) column and eluted at 0.15 ml/min using a constant 10 mM ammonium formate pH 9.0 and a multistep gradient from 10% to 50% acetonitrile over 21 min, then 80% for 1 min and held at 80% for another minute, followed by re-equilibration. Overall, 24 fractions were collected at 35 s intervals from 10 to 24 min. Collection was delayed for 9 min to prevent collection of unreacted TMTpro. In total, 24 fractions were concatenated into 12 and dried before being loaded onto C18 stage tips.

After fractionation, for TMT10plex experiments, ~5% of each fraction was reserved as the non-phospho-enriched ("N") sample and the remaining 95% was subjected to phospho-peptide enrichment ("PE") using magnetic Ti4+ IMAC beads (MagReSyn) as described above. Eluted peptides were dried and then loaded onto C18 stage tips.

## Mass spectrometry

Peptides were eluted from stage tips in 40 μL of 80% acetonitrile in 0.1% TFA and concentrated down to 1 μL by vacuum centrifugation (Concentrator 5301, Eppendorf, UK). They were then prepared for LC-MS3 analysis by diluting to 5 μL by 0.1% TFA. All fractions from every TMT experiment were injected on Orbitrap Fusion™ Lumos™ Tribrid™ mass spectrometer. coupled online, to an Ultimate 3000 HPLC (Dionex, ThermoFisher Scientific, UK). Peptides were separated on a 50 cm (2-μm particle size) EASY-Spray column (Thermo Scientific, UK), which was assembled on an EASY-Spray source (Thermo Scientific, UK) and operated constantly at 50 °C. Mobile phase A consisted of 0.1% formic acid in LC-MS grade water and mobile phase B consisted of 80% acetonitrile and 0.1% formic acid. Peptides were loaded onto the column at a flow rate of 0.3 μL min⁻¹ and eluted at a flow rate of 0.25 μL min⁻¹ according to the following gradient: 2–40% mobile phase B in 120 min and then to 95% in 11 min. Mobile phase B was retained at 95% for 5 min and returned back to 2% a minute after until the end of the run (160 min). IP-MS experiments were injected on an Orbitrap Exploris™ 480 mass spectrometer, under the same conditions with a gradient 30 min longer (total run time 190 min).

Survey scans were recorded at 120,000 resolution (scan range 380–1500 $m/z$) with an ion target of 4.0e5, and injection time of 50 ms. MS2 was performed in the ion trap at a turbo scan mode, with ion target of 2.0E4 and CID fragmentation with normalised collision energy of 35, Q activation parameter at 0.25 and CID activation time at 10 ms. For the TMT16plex experiment, MS2 was done in the ion trap at rapid scan mode, and HCD fragmentation (Olsen et al, 2007) with normalized collision energy of 28, was used. The isolation window in the quadrupole was 0.7 Thomson. Only ions with charge between 2 and 7 were selected for MS2. Dynamic exclusion was set at 70 s. MS3 scans were performed in the orbitrap, with the number SPS precursors set to 5. The isolation window was set at 2, and the resolution at 50,000 with scan range 100–150 $m/z$. HCD fragmentation was performed with normalised collision energy of 65 (55 for 16plex). For the IP-MS, MS1 scans (350–1500 $m/z$) were recorded at 120,000 resolution with RF lens set at 40. MS2 resolution was set to 15,000, with isolation window of 1.4 Thomson and normalised collision energy of 30.

All proteomics data associated with this study have been deposited to the ProteomeXchange Consortium via the PRIDE partner repository (Perez-Riverol et al, 2022) with the dataset identifier PXD048618.

## MaxQuant conditions

All four TMT10 timecourse experiments, including "N" and "PE" samples and 24 fraction samples total per TMT10 experiment, were analysed together in a single MaxQuant (MQ) (Cox and Mann, 2008) session. For the metaphase I arrest experiment, "N" and "PE" samples were also analysed in the same MQ session. The "N" and "PE" samples of the TMT16 experiment were analysed separately. The version 1.5.3.30 was used to process the raw files, and search was conducted against our in-house *Saccharomyces cerevisiae*

complete/reference proteome database (released from SGD in September 2019), using the Andromeda search engine (Cox et al, 2011). For the IP-MS experiment, we used version 1.6.1.0 with the same general parameters. For the first search, peptide tolerance was set to 20 ppm while for the main search peptide tolerance was set to 4.5 pm. Isotope mass tolerance was 2 ppm, and maximum charge to 7. Digestion mode was set to specific with trypsin allowing maximum of two missed cleavages. Carbamidomethylation of cysteine was set as fixed modification. Oxidation of methionine and phosphorylation of serine, threonine and tyrosine were set as variable modifications. For quantification, we set "Reporter Ion MS3", and we chose the 10plex labels (16plex for the 16plex experiment) as provided by MQ. We applied the correction factors provided by the manufacturer. For the IP-MS experiment, Label-free quantitation analysis was performed by employing the MaxLFQ algorithm as described (Cox et al, 2014). Absolute protein quantification was performed as described (Schwanhäusser et al, 2011). Peptide and protein identifications were filtered to 1% FDR.

## Data analysis in R

All data analysis was performed using the proteinGroups.txt and Phospho(STY)Sites.txt files from MaxQuant using R (v 4.2.3) within the RStudio environment. All scripts used for analysis are provided in ComputerCode EV1. The R scripts are also available on Github at https://github.com/lori-koch/meiotic-divisions-phosphoproteomics. Common yeast gene names were added to the tables by matching the systematic OLN to the common gene names found in the table from https://ftp.uniprot.org/pub/databases/uniprot/knowledgebase/complete/docs/yeast.txt.

## TMT data analysis

For proteins, the "Reporter.intensity.corrected." columns in the proteinGroups table corresponding to the measurements from the non-phospho-enriched ("N") samples were analysed. For phospho-sites, the "Reporter.intensity.corrected" columns in the Phospho(STY)Sites table corresponding to the measurements from the phospho-enriched ("PE") samples were analysed. Only singly phosphorylated site intensities (i.e. Reporter.intensity.corrected columns with suffix "___1") were analysed.

Proteins or sites marked as contaminants or reverse-matching by MQ were filtered out. Missing values (zeros) were changed to NAs so they were not considered during normalisation. After normalisation, missing values were converted back to true zeros. To normalise between TMT reporter channels, all values in each reporter column were multiplied by a scaling factor consisting of the median intensity of all reporter columns divided by the median reporter intensity of that individual column. For phospho-site normalisation to the protein level, the phospho-site intensities from the "PE" reporter columns in the Phospho(STY)Sites table were divided by the corresponding reporter intensity of the same protein from the "N" reporter columns from the proteinGroups table and multiplied by 1000.

## Analysis of TMT10 prophase block-release timecourses

Normalised protein and phospho-site intensities were $\log_2$ transformed and batch effect correction carried out using the

limma package removeBatchEffect function. Following this, $\log_2$ transformation was reversed by antilog 2. NA values were changed back to true zeros. All subsequent analysis was carried out using these median-normalised and batch-corrected intensities.

To be considered part of the final wild-type or *spo13Δ* timecourse dataset, proteins were required to be detected, having a reporter intensity value >0, in at least one timepoint of both of the two biological replicates of the timecourse. For phospho-sites, the phospho-site was required to have >0 reporter intensity in at least two sequential timepoints in both of the biological replicate timecourses and have a localisation probability >0.75. Only phospho-sites that could be normalised to the corresponding protein level in all 10 timepoints were analysed further. To be considered part of the final combined wild-type and *spo13Δ* timecourse dataset, proteins and phospho-sites were required to meet thresholds for both individual strain datasets.

To be called significantly dynamic from time 0, a protein or phospho-site needed to meet a fold change and t-test p-value cutoff that considered the abundances in both of the two TMT10 biological replicates of the timecourse. 9 pairwise comparisons between time 0 and each subsequent timepoint were considered. The protein or phospho-site needed to meet the following conditions: First, for at least 1 of the 9 comparisons, it must have a median fold change of greater than 1.5 or less than 1/(1.5). Second, for the same timepoint comparison that meets the fold change cutoff, the difference in abundance needed to be considered significant and consistent between the replicates by returning *t* test *P* value of <0.05. The t.test() function from the stats package was used with default parameters, running a Welch's *t* test.

To be considered significantly different abundance between strains, protein or phospho-site abundances at matched timepoints were compared (i.e. t45 abundance in wild-type vs t45 abundance in *spo13Δ*) and the same fold change >1.5 or <1/(1.5) and *t* test *P* value < 0.05 thresholds were applied.

Proteins or phospho-sites which had 0 intensity at one timepoint and >0 intensity at the other timepoint to be compared were allowed to meet the fold change cutoff for being called significantly dynamic from time zero or significantly different between strains. In all, 7% of the wild-type proteins and 4% of the combined wild-type and *spo13Δ* protein dataset were considered significantly dynamic from time zero based on a 0 vs >0 comparison and a *t* test *P* value < 0.05. 29% of wild-type phospho-sites and 17% of the combined wild-type and *spo13Δ* phospho-sites were considered dynamic from time zero based on a 0 vs >0 comparison and a *t* test *P* value < 0.05. 16% of proteins and 23% of phospho-sites were called significantly different between wild-type and *spo13Δ* based on 0 vs >0 comparisons as well as a *t* test *P* value < 0.05. Fold changes from 0 vs >0 comparisons, which are Inf, were imputed by replacing the zeros with the minimum value in the corresponding total dataset and re-calculating the median fold changes. The median-normalised, batch-corrected intensities for each of the two biological replicates of each strain timecourse were averaged together for downstream analyses. This means that the intensities of proteins or phospho-sites which had zero intensity at one timepoint in one replicate were reduced by half.

In the metaphase-to-anaphase analysis heatmaps, fold changes outside the bounds of the central colour scale were "trimmed" to facilitate seeing the trend in the central data; Fold changes >2.2 were trimmed to 2.2 and fold changes <0.2 were trimmed to 0.2. In the metaphase-to-anaphase GO term analysis barplots, if a protein

increased after the metaphase timepoint, the *P* value was plotted as +log10(*P* value). However, if the protein decreased after metaphase, the −log10(*P* value) was plotted.

For the metaphase-to-anaphase protein and phospho-site analysis, metaphase I was defined as 75 min and metaphase II as 120 min. Because 135 min was the peak of anaphase II according to the spindle immunofluorescence and phospho-Net1 profile (Fig. 1A,E), the timepoint just before that, 120 min, was considered metaphase II. Proteins or phospho-sites were called significantly changing from metaphase if they reached >1.5 or < 1/(1.5) fold change from the metaphase timepoint and were significantly different abundance from the metaphase timepoint by *t* test *P* value < 0.05.

For the metaphase-to-anaphase phosphosite analysis, metaphase I-scaled values for each phospho-site were calculated by dividing all 10 timepoint intensities by the intensity at the 75 min timepoint. Metaphase II scaled intensities for each phospho-site were calculated by dividing all 10 timepoint intensities by the intensity at the 120 min timepoint. In the lineplots, the median and 25–75% quartile range of the metaphase scaled intensities of motif-matching sites are plotted.

For all other lineplots, "scaled intensity" means scaled to the mean. To be mean-scaled, it means that each unique protein or phospho-site was divided by its own mean intensity of all 10 timepoint values, including zeros. For the wild-type and *spo13Δ* combined dataset lineplots, protein or phospho-sites were scaled to the mean of both the wild-type and *spo13Δ* timecourses together, allowing for inter-strain differences in abundance to be preserved. For summary lineplots (referring to more than one protein or phospho-site), the median and 25–75% quartile range of the mean-scaled intensities are plotted.

### Analogue-sensitive kinase experiments analysis

For the *cdc28-as* experiment, to categorise phospho-sites into groups that were not changed, decreased or increased between *cdc28-as* and wild type, only sites that were detected in 1 out of 2 replicates (with normalised intensity >0) were considered. A fold change (*cdc28-as*/wild type) of 1.5 was used as the arbitrary cutoff for significance. Sites that had non-finite median fold changes (due to the median fold change having a zero denominator due to allowing sites detected in 1 out of 2 reps) were categorised as "no change".

For the *cdc5-as* experiment, to categorise phospho-sites into groups that were not changed, decreased or increased between *cdc5-as* and wild type, only sites that were detected in 2 out of 2 replicates (with normalised intensity >0) were considered. A fold change (*cdc5-as*/wild type) of 1.5 was used as the arbitrary cutoff for significance.

### Metaphase I arrest TMT10 experiment analysis

Only proteins with >0 normalised reporter intensity in at least three replicates of all three strains were considered part of the final dataset. Only phospho-sites with >0 normalised reporter intensity in at least three replicates of all three strains and which had a localisation probability >0.75 were considered part of the final dataset. Significant differences in abundance between proteins and phospho-sites between strains were called according to a fold change and *t* test P-value cutoff similar to the timecourse experiments. The protein or phospho-site needed to vary between the two strains by a median fold change >1.5 or <1/(1.5), considering all 3 or 4 replicates of each strain. Additionally, the difference in abundance between the two strains, considering all replicates, needed to return a *t* test *P* value < 0.05. The base R t.test() function was used with default parameters, running a Welch's *t* test.

### Hierarchical clustering

Hierarchical clustering was performed using hclust function and cutree functions from the stats package along with the pheatmap package for visualisation. Ward's method was used in the hclust function (method = "ward.D2"). The number of clusters k was determined by identifying the inflection point in a plot of within groups sum of squares vs the number of clusters (ie the 'elbow method').

### GO term enrichment analysis

GO term enrichment analysis was performed using the gprofiler2 package with organism = 'scerevisiae' and otherwise default settings. This runs a hypergeometric test followed by correction for multiple testing. GO terms were considered significantly enriched if the *P* value < 0.05.

### Phospho-site motif logo analysis

The ggseqlogo package was used to make standard sequence logos (Wagih, 2017). The IceLogo software (Colaert et al, 2009) was used to make IceLogos using a custom background list of sequences. The percent difference between the two sequences was plotted, with the *P* value cutoff set to 0.05. Kinase consensus motifs were all based on those reported in (Mok et al, 2010), with the exception of the motif for Mec1[ATR] defined in (Kim et al, 1999) and the motif for Ime2 reported in (Holt et al, 2007).

### Motif analysis

Protein sequences of proteins that had significantly decreased or "no change" phosphorylation in *spo13Δ* or *spo13-m2* vs wild-type were downloaded from SGD YeastMine. Whether each protein contained an S[ST]P motif at any position was determined by searching for this motif using a regular expression.

### Fisher tests

Fisher tests were run via the fisher.test() function from the stats package. In all figure panels showing Fisher test results, a Fisher test was carried out between the pairs of groups in all the barplots shown, however if there the *P* value was not <0.05, a bar was not shown as the difference was not significant. In Figs. 3 and EV2, Fisher tests were performed for the enrichment of all 12 kinase motifs in all clusters but only enrichments or depletions with *P* values < 0.05 were starred. *P* values < 0.001 were considered significant with three asterisks, *P* values < 0.01 were considered significant with two asterisks, and *P* values < 0.05 were considered significant with one asterisk.

For the Fisher tests on clustering results in Figs. 3, EV2, and 8, the number of motif-matching sites in selected clusters was compared with the number of motif-matching sites in all clusters including the selected cluster (total). For the Fisher tests accompanying the IceLogo analyses, the number of motif-matching sites in the selected non-overlapping subsets were compared, to match the lists used to make the IceLogos.

## Cdc5 IP-LFQMS analysis

Results from the Cdc5 immunoprecipitation experiment were analysed using the Differential Expression of Proteins ('DEP') R package (Zhang et al, 2018). Abundances were normalised using the normalize_vsn() function and missing values were imputed with the impute() function, with the fun argument set to "MinProb" and the $q$ value = 0.01.

# Data availability

The datasets and computer code produced in this study are available in the following databases: Proteomics, Phosphoproteomics and Immunoprecipitation-mass spectrometry data PRIDE: PXD048618 (http://www.ebi.ac.uk/pride). Computer scripts for analysis of proteomics data Github: https://github.com/lori-koch/meiotic-divisions-phospho-proteomics.

# Peer review information

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

## Acknowledgements

The authors gratefully acknowledge the Wellcome Discovery Research Platform for Hidden Cell Biology Proteomics Core for mass spectrometry support. We are grateful to Georg Kustatcher for helpful discussions, to Shaun Webb for bioinformatics support and advice, and Lucia Massari, Gerard Pieper and Menglu Wang for comments on the manuscript. We are grateful to Tiasha Ghosh, Marina Hamaia, and Lucy Munro for their feedback on the web-based visualisation tool. We are grateful to Sandra Touati and Katja Wassmann, and to Joao Matos for sharing results prior to publication. This work was funded through a Wellcome Investigator award to ALM [220780], a Wellcome and Royal Society Sir Henry Dale Fellowship to TL [206211], a Wellcome Multi-User Equipment Grant [218305], core funding for the Wellcome Centre for Cell Biology [203149] and a Wellcome Discovery Research Platform Award [226791].

## Author contributions

**Lori B Koch**: Conceptualisation; Data curation; Software; Formal analysis; Investigation; Visualisation; Methodology; Writing—original draft; Writing—review and editing. **Christos Spanos**: Data curation; Formal analysis; Investigation; Methodology; Writing—review and editing. **Van Kelly**: Investigation; Methodology; Writing—review and editing. **Tony Ly**: Supervision; Funding acquisition; Writing—review and editing. **Adele L Marston**: Conceptualisation; Supervision; Funding acquisition; Writing—original draft; Writing—review and editing.

## Disclosure and competing interests statement

The authors declare no competing interests.

# Expanded View Figures

**Figure EV1. Protein dynamics at the metaphase-to-anaphase transition in meiosis I and II.**

(**A**) Fold change in protein abundance of proteins that significantly change from metaphase I to anaphase I. (**B**) Fold change in protein abundance of proteins that significantly change from metaphase II to anaphase II. (**C**) GO term analysis of proteins from (**A**). Data information: Statistics: Cumulative hypergeometric test followed by correction for multiple testing (gprofiler2 R package gost function default settings). (**D**) GO term analysis of proteins from (**B**). Data information: Statistics: Cumulative hypergeometric test followed by correction for multiple testing (gprofiler2 R package gost function default settings). Source data are available online for this figure.

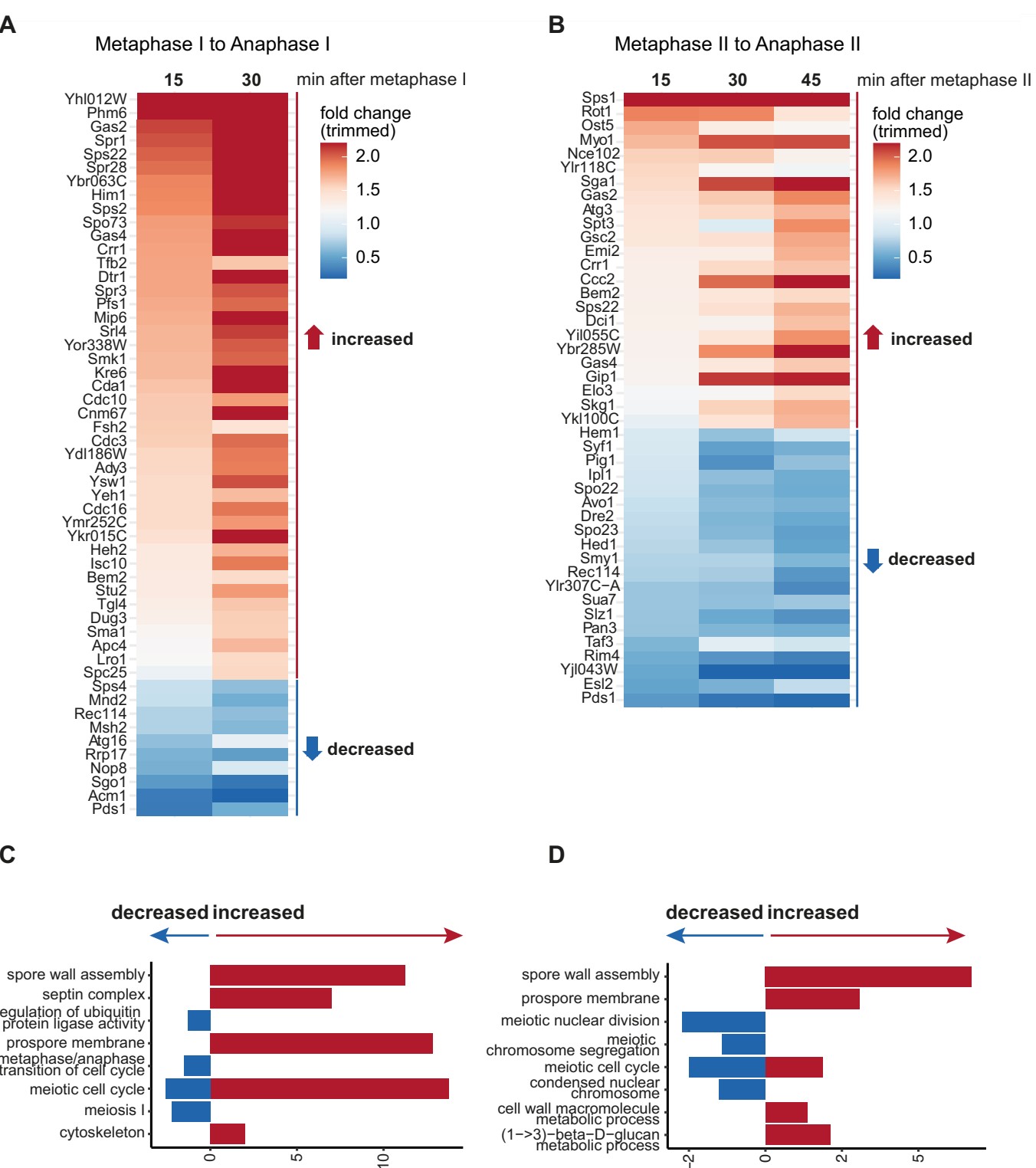

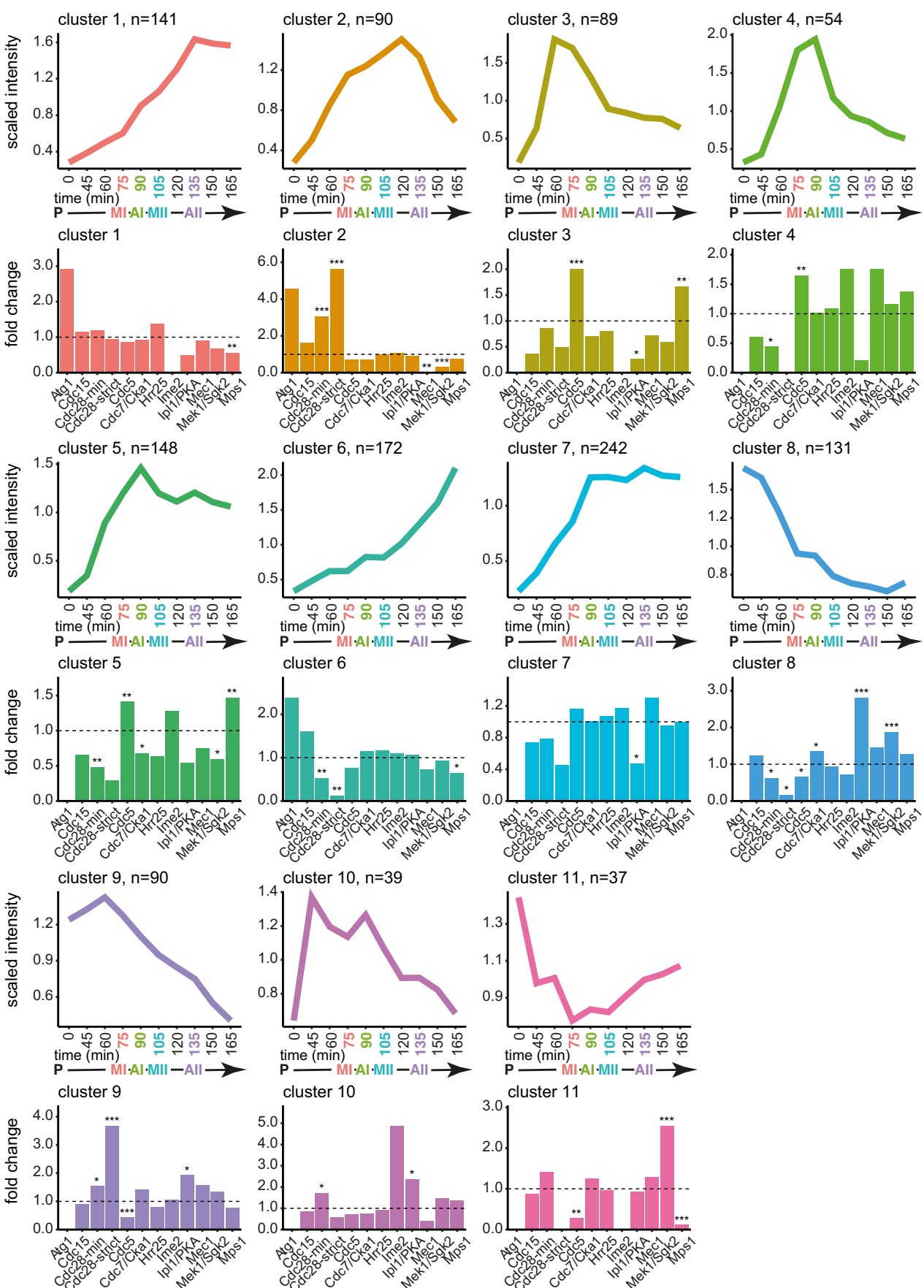

**Figure EV2.  Enrichment of kinase consensus motifs for clusters of dynamic phosphorylation sites.**

Median lineplots of all clusters from Fig. 3A and kinase motif enrichment analysis bar graphs. Asterisks represent p value from Fisher's exact test **$P < 0.01$; *$P < 0.05$). Source data are available online for this figure.

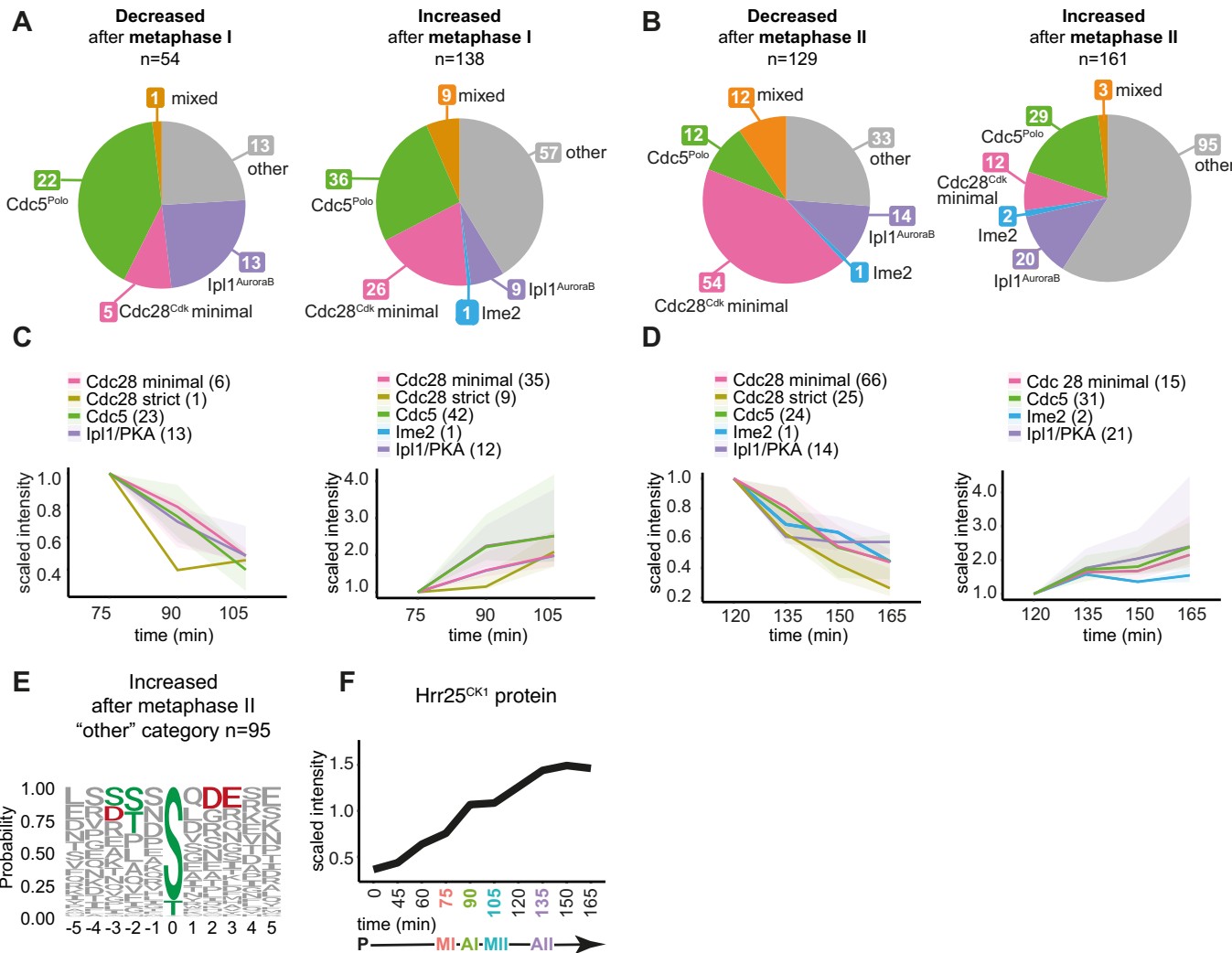

**Figure EV3.  Phosphorylation dynamics at the metaphase-to-anaphase transitions in meiosis I and II.**

(**A**) Motifs matching phospho-sites that significantly decrease (left) or increase (right) at the metaphase I to anaphase I transition. (**B**) Motifs matching phospho-sites that significantly decrease (left) or increase (right) at the metaphase II to anaphase II transition. (**C**) Median change of motif-matching phospho-sites decreasing (left) or increasing (right) from metaphase I to anaphase I. Abundance scaled to 75 min (metaphase I). (**D**) Median change of motif-matching phospho-sites decreasing (left) or increasing (right) from metaphase II to anaphase II. Abundance scaled to 120 min (metaphase II). (**E**) Motif logo of phospho-sites that are increased after metaphase II from (**B**) (right), which do not match any of the selected motifs, from the "other" category $n = 95$. (**F**) Abundance of Hrr25^CK1 protein rises in meiosis II. Source data are available online for this figure.

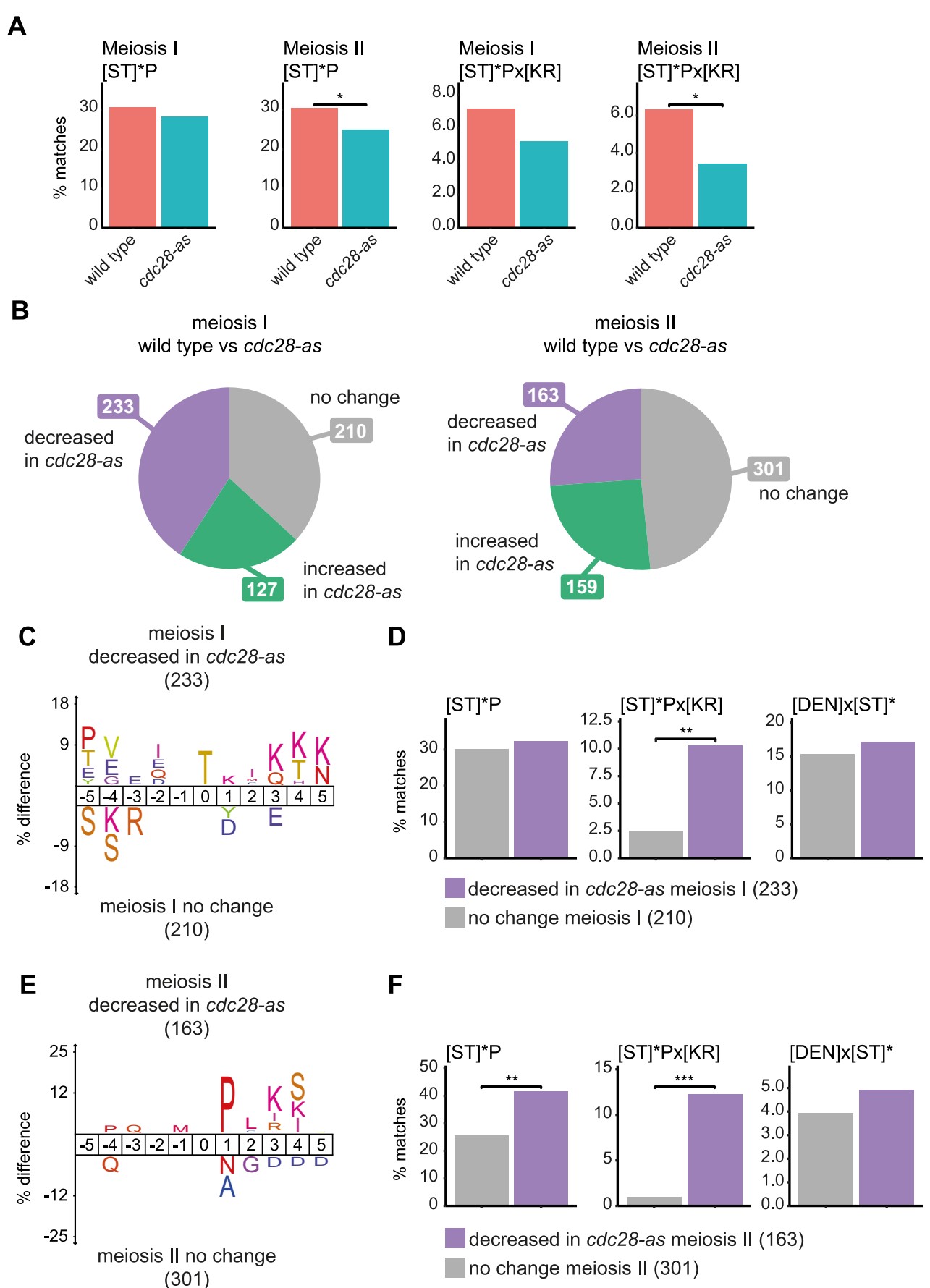

◀ **Figure EV4. Strict Cdk consensus site phosphorylation is the best predictor of Cdc28[Cdk1] kinase activity.**

(A) Fisher tests comparing the frequency of matching the indicated motifs among all sites detected in the indicated samples. Data information: Statistics: Fisher's exact test, *$P < 0.05$. (B) Analysing only phospho-sites detected in both *cdc28-as* and wild-type, pie charts of the proportion of no change, increased or decreased sites when comparing *cdc28-as* and wild type in either the meiosis I samples (left) or meiosis II samples (right). (C) Icelogo comparing phospho-sites decreased in *cdc28-as* vs wild type in meiosis I and sites that are not significantly changed. (D) Fisher tests comparing the enrichment of motifs in the indicated groups of phospho-sites (same groups as in pie chart in (B, left)). Data information: Statistics: Fisher's exact test, **$P < 0.01$. (E) Icelogo comparing phospho-sites decreased in *cdc28-as* vs wild type in meiosis II and sites that are not significantly changed. (F) Fisher tests comparing the enrichment of motifs in the indicated groups of phospho-sites (same groups as in pie chart in (B, right)). Data information: Statistics: Fisher's exact test, **$P < 0.01$, ***$P < 0.001$. Source data are available online for this figure.

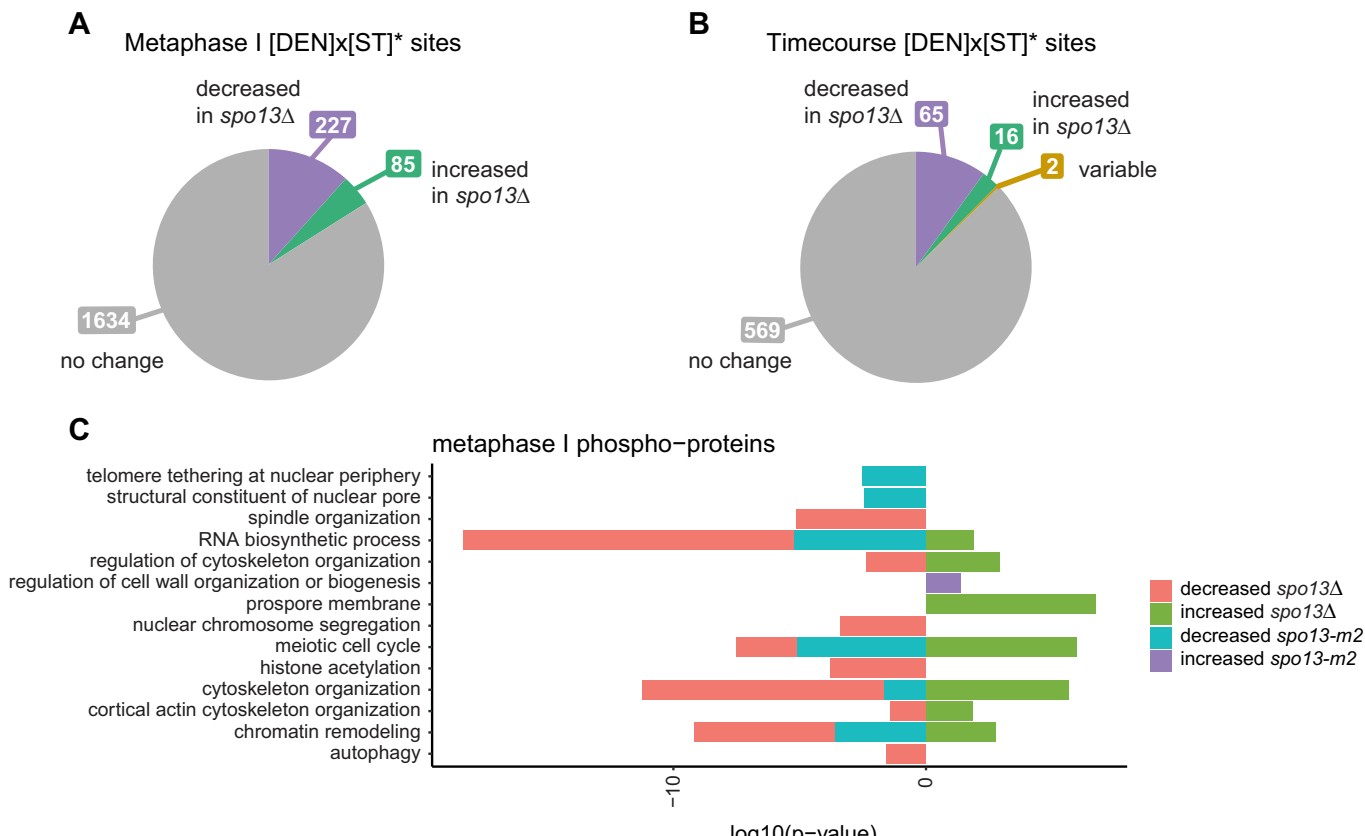

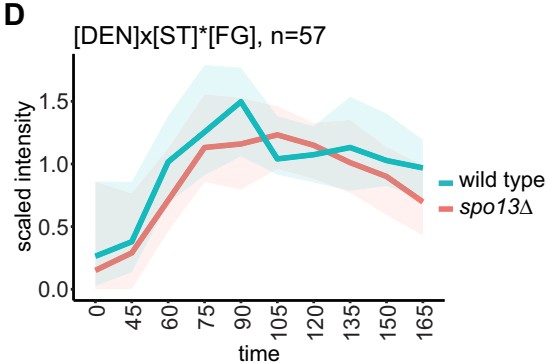

**Figure EV5. Comparison of Cdc5$^{Polo}$ kinase motif phosphorylation between wild type and *spo13Δ*.**

(A) Proportion of [DEN]x[ST]* motif-matching phospho-sites with significantly different abundance in *spo13Δ* versus wild type in metaphase I-arrested cells. (B) Proportion of [DEN]x[ST]* motif-matching phospho-sites with significantly different abundance in *spo13Δ* versus wild type in the meiotic timecourse experiments. (C) GO terms enriched among proteins with significantly different phosphorylation in *spo13Δ* or *spo13-m2* versus wild type. Data information: Statistics: Cumulative hypergeometric test followed by correction for multiple testing (gprofiler2 R package gost function default settings). (D) Abundance of phospho-sites matching the [DEN] x[ST]*[FG] motif among sites detected in both replicates of wild type and *spo13Δ* across the timecourse. Source data are available online for this figure.

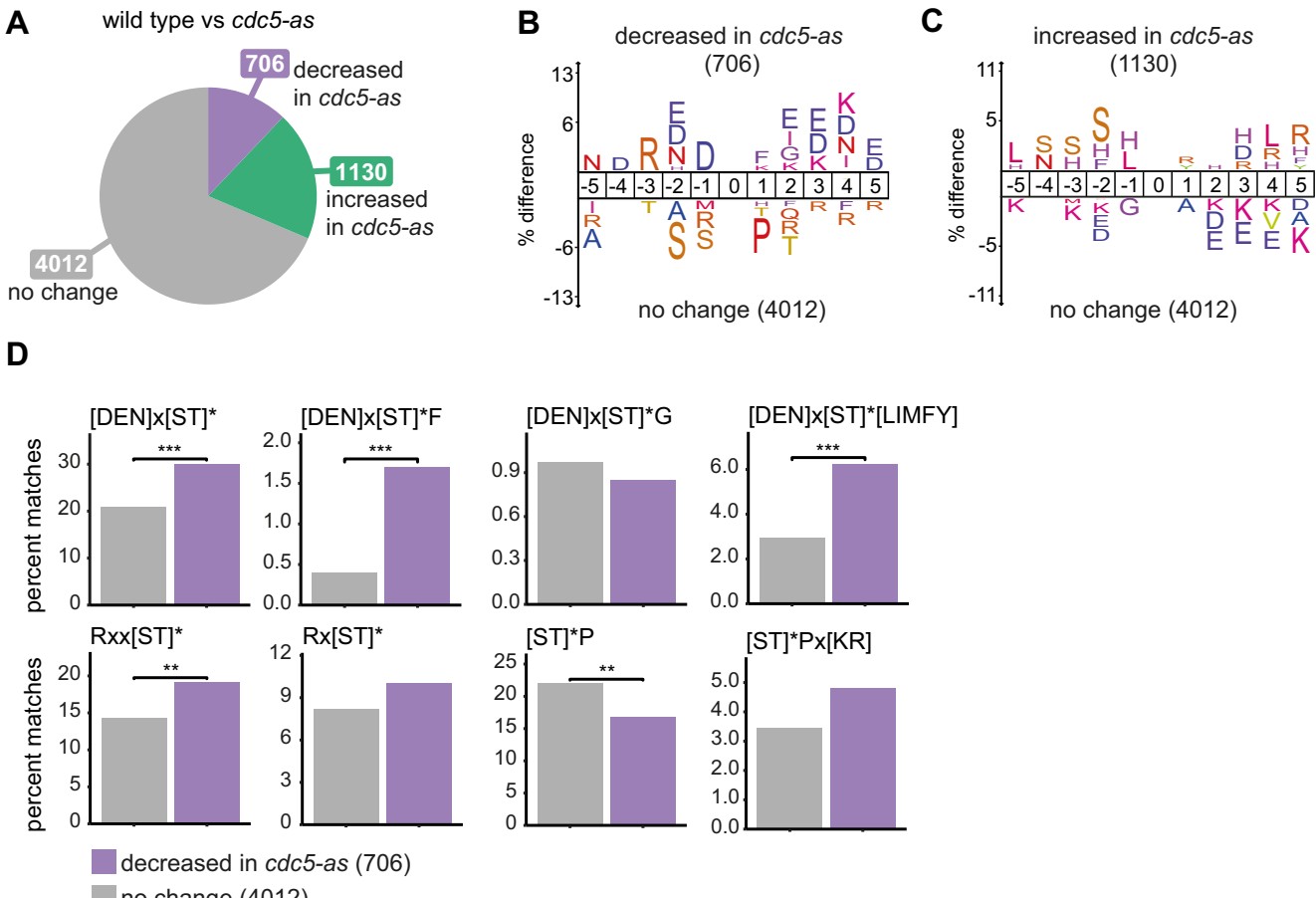

**Figure EV6.** **[DEN]x[ST]\* and [DEN]x[ST]\*F motif phosphorylation depends on Cdc5<sup>Polo</sup>.**

(A) Pie chart showing the proportion of increased, decreased and no change sites between *cdc5-as* and wild type in prometaphase. (B) IceLogo comparing amino acid frequency between sites that were decreased in *cdc5-as* compared to no change sites. (C) IceLogo comparing amino acid frequency between sites that were increased in *cdc5-as* compared to no change sites. (D) Fisher tests comparing the number of the indicated motif-matching sites between the groups of sites that were decreased in *cdc5-as* (purple) or no change (grey). Data information: Statistics: Fisher's exact test, \*\**P* < 0.01, \*\*\**P* < 0.001. Source data are available online for this figure.

