## [Peer Review File · The EMBO Journal]

Rewiring of the phosphoproteome executes two meiotic divisions in budding yeast

Lori Koch, Christos Spanos, Van Kelly, Tony Ly, and Adele Marston

Corresponding author(s): Adele Marston (adele.marston@ed.ac.uk)

Review Timeline:

Submission Date:	13th Sep 23
Editorial Decision:	19th Oct 23
Revision Received:	19th Jan 24
Editorial Decision:	31st Jan 24
Revision Received:	6th Feb 24
Accepted:	7th Feb 24

Editor: Hartmut Vodermaier

Transaction Report:

Prof. Adele L Marston
University of Edinburgh
Wellcome Centre for Cell Biology
Max Born Crescent
Edinburgh EH9 3BF
United Kingdom

19th Oct 2023

Re: EMBOJ-2023-115604
Rewiring of the phosphoproteome executes two meiotic divisions

Dear Adele,

Thank you for submitting your study on meiotic phosphoproteome rewiring for our consideration. I sent it to three expert referees, who have now returned their comments, copied below. The reviewers appreciate the extent and value of the new datasets, but as you will see, referees 1 and 2 are not yet fully convinced by all interpretations and conclusions, and indicate potential alternative explanations that have not been decisively ruled out.

Should you be able to satisfactorily address the various specific concerns raised by the reviewers, we would be happy to pursue a revised manuscript further for EMBO Journal publication. Since we generally consider only a single round of major revision, I would in this case encourage you to get back to me with a tentative point-by-point response and revision plan already during the early stages of the revision, so that we could discuss the key revision requirements and how the main concerns may be best addressed - in particular referee 1's request for direct mechanistic testing of some conclusions, and referee 2's related comment on the importance of direct biochemical demonstration of the connection between certain sites and particular kinases. We could also offer extension of the default three-months revision period if needed, with our 'scooping protection' (meaning that competing work appearing elsewhere in the meantime will not affect our considerations of your study) remaining of course valid also throughout this extension.

Detailed information on preparing, formatting and uploading a revised manuscript can be found below and in our Guide to Authors. Thank you again for the opportunity to consider this work for The EMBO Journal, and I look forward to hearing from you in due time.

With kind regards,

Hartmut

3) Revised manuscript text (including main tables, and figure legends for main and EV figures) has to be submitted as editable

text file (e.g., .docx format). We encourage highlighting of changes (e.g., via text color) for the referees' reference.

4) Each main and each Expanded View (EV) figure should be uploaded as individual production-quality files (preferably in .eps, .tif, .jpg formats). For suggestions on figure preparation/layout, please refer to our Figure Preparation Guidelines:

8) Please note that supplementary information at EMBO Press has been superseded by the 'Expanded View' for inclusion of additional figures, tables, movies or datasets; with up to five EV Figures being typeset and directly accessible in the HTML version of the article. For details and guidance, please refer to:

embopress.org/page/journal/14602075/authorguide#expandedview

9) Digital image enhancement is acceptable practice, as long as it accurately represents the original data and conforms to community standards. If a figure has been subjected to significant electronic manipulation, this must be clearly noted in the figure legend and/or the 'Materials and Methods' section. The editors reserve the right to request original versions of figures and the original images that were used to assemble the figure. Finally, we generally encourage uploading of numerical as well as gel/blot image source data; for details see: embopress.org/page/journal/14602075/authorguide#sourcedata

At EMBO Press, we ask authors to provide source data for the main manuscript figures. Our source data coordinator will contact you to discuss which figure panels we would need source data for and will also provide you with helpful tips on how to upload and organize the files.

Further information is available in our Guide For Authors:

In the interest of ensuring the conceptual advance provided by the work, we recommend submitting a revision within 3 months (17th Jan 2024). Please discuss the revision progress ahead of this time with the editor if you require more time to complete the revisions. Use the link below to submit your revision:

Link Not Available

Referee #1:

The manuscript by Martson and colleagues describes a study where MS-based phospho-proteomics analysis was used to study phosphorylation events associated with Meiosis I and Meiosis II in budding yeast. The authors raised two general key questions, (1) how is the cell cycle machinery remodelled to direct the unique pattern of segregation in meiosis I, (2) what ensures that meiosis I is followed by another M phase, meiosis II, rather than S phase as in the canonical cell cycle? They observed distinct waves of phosphorylation signatures across the meiotic divisions and identified 1,233 significantly dynamic phosphorylation sites of which the majority (78%) showed increased abundance after prophase, while ~21% decreased. Interestingly, they found a relatively large cluster of sites with the consensus site sequence fitting the one for Polo kinase Cdc5 in Meiosis I, while the CDK sites showed a somewhat biphasic profile in MI and MII. They also found that the effect of a deletion of, or a mutation in Spo13, a member of a Meikin family of meiosis-specific master regulators influences the substrate site selection by Polo-Cdc5, with enrichment of the Polo kinase consensus motif sub-version with a phenylalanine at position +1. The Spo13Meikin binds the Polo-box of Cdc5 and the authors proposed that this interaction somehow must modulate structural features of the active site region in the Cdc5 kinase and introduce the meiosis-specific (+1F) change to the substrate consensus phosphorylation motif.

The manuscript is well-written and the study is technically well performed. The findings of temporal waves of phosphorylation events (at Polo kinase, basophilic and CDK motifs) and the Spo13Meikin-aided meiosis-specific substrate targeting are important contributions to the field. However, the claim of the discovery of a new meiosis-specific consensus sequence for Polo-Cdc5-Spo13Meikin complex is too premature and requires more experimental proof. Both biochemical and in vivo evidence are

needed for the claim that Spo13Meikin "re-wires the phosphoproteome" that could explain the roles of Spo13Meikin in sister kinetochore mono-orientation, cohesion protection and the execution of two meiotic divisions.

Specific points:

- 1) The authors suggest that phosphatases active at the meiosis I to II transition could have a preference for Cdc28Cdk1 sites with lysine or arginine in the +2 position, while sites with a different residue at +2 are protected from dephosphorylation. The authors likely meant +3, not +2. However, the main point is that the evidence for this suggestion has very little experimental backing. Are the authors sure that the majority of minimal S/TP sites without the lysin or arginine, that they use as a control group, are really phosphorylated by CDK and not by other kinases with similar +1P specificity?
- 2) Direct mechanistic studies are required to test these ideas and also to understand how "Cdks are permitted to rise again for entry into meiosis II". Also, replication factors for licensing have many +3K/R CDK consensus sites. How the model of drop of optimal site phosphorylation would provide maximal protection against the re-licensing of origins between MI and MII? Can the cyclin quantitative profile parallel to the optimal site drop (Fig 5F) be measured? Which cyclins are expressed and degraded?
- 3) The authors suggest that Cdc5Polo uses its Polo Binding Domain (PBD) to bind to phosphorylated Spo13Meikin in the same way it normally binds to substrates, and since the usual mode of substrate binding is blocked, Spo13-Cdc5Polo must therefore use another mechanism to target substrates. They propose that potentially, Spo13Meikin itself targets Spo13Meikin-Cdc5Polo to substrates. It is hard to imagine without any biochemical or structural data how the preference for phenylalanine (F) or glycine (G) in the +1 position can be gained via such a mechanism. Highly unbelievable and pure speculation. It could be that polo-box docking just broadens the specificity and shielding this docking via Meikin would amplify the +1F-containing optimal consensus sites. There could be other explanations. It is a too bold statement that by redirecting Cdc5Polo to specific substrates containing the [DEN]_x[ST]*F motif at metaphase I, Spo13Meikin establishes the meiotic programme. The subtitle of the section certainly cannot be "Identification of a Spo13Meikin - Cdc5Polo consensus motif".
- 4) Fig 6 C, d and E, F are not clearly described. What are these sequence logos really showing? Also, how do the authors explain the effect of Spo13 deletion on CDK and Ipl1 (basophilic) kinase effects?
- 5) Page 10-11 Cluster 10 showed a peak of phosphorylation in meiosis I in wild type and a delay in phosphorylation in spo13D and was enriched for the strict Cdc28Cdk motif [ST]*_x[KR] (Fig 5C). The +1Pro is missing?

Referee #2:

This manuscript from Koch et al describes a mass spectrometry-based investigation of changes in the yeast phosphoproteome during meiosis. The authors isolate samples across a well-synchronized meiotic time course and at each time point analyze the abundance of both total protein and phosphopeptides. The authors identify >1000 phosphosites that are dynamic across the time course. These data were then analyzed in two ways: first, phosphosites were clustered, sorting them into 11 different patterns of behavior showing enrichment or loss at specific times in meiosis. Second, the regions around the phosphoacceptor in each peptide were compared to established consensus sequences for different kinases known to be involved in meiosis and spore formation. This analysis correlates particular clusters with specific kinases and, therefore, implicates their activity at specific points in meiosis. It reveals, among other things, two waves of phosphorylation matching the CDK consensus site as well as two waves matching the Cdc5 consensus.

The authors then apply the same protocol to analyze spo13 mutant cells and compare them to the wild-type. Spo13 is known to bind to the Cdc5 kinase and is required for cells to undergo two successive chromosome divisions during meiosis. This analysis revealed about three hundred phosphosites whose abundance was altered in the spo13 mutant cells. Applying the consensus motifs to these sites revealed not only that the Cdc5 consensus was over-represented in the set of decreased phosphosites, but that other kinase consensus sites were also affected, suggesting that these kinases are regulated by Spo13.

Finally the authors specifically compared phosphopeptides from WT, spo13Δ and spo13-m2 (an allele that does not bind Cdc5) cells arrested at metaphase I, when Spo13 is thought to be most active. The phosphosites reduced in these conditions reveal not just a reduction in Cdc5 consensus sites, but specifically a slightly modified Cdc5 consensus, suggesting that this represents a favored site of Spo13-Cdc5 phosphorylation.

This is a very thorough study that generates significant new hypotheses about the regulator wiring of meiotic kinases. The timing

of the increase or decrease of phosphorylation on consensus site of known kinases largely matches expectations and this gives confidence that the newer insights are correct.

A weakness of this study is that the analysis is based entirely on consensus site and motif frequency. Many different kinases fall into the category of acid/base/proline directed and so the appearance of a particular phosphosite within a consensus can't be taken as firm evidence that the phosphorylation is performed by a particular kinase. In some cases, the conclusions are weaker than others. In particular, the inference that Spo13 regulates Hrr25 seems quite speculative and should be qualified in the text.

The work would have been strengthened by direct biochemical demonstration of the connection between some of the sites and particular kinases. For example, testing whether Spo13-Cdc5 has a different phosphosite preference than Cdc5 alone. However, this a general issue with all 'omic level analyses and, this issue notwithstanding, the study is very well done and generates not only a large body of information that will be useful to the field, but also specific models and hypotheses for further, mechanistic analysis.

Other comments:

In many cases, eg. the graphs for clusters 1, 2, and 4 in Figure 5B, there does not appear to be any difference between the median intensity curves for WT and spo13 Δ . This seems surprising given that the clusters have been formed from sites that "significantly vary" between the two strains.

It would be useful in the supplementary data to indicate which phosphosites fell into which consensus so that readers could see which sites on which proteins are implicated as specific targets of the different kinases.

Minor points

Figure EV5 legend "G" should be "F"

Referee #3:

Koch et al undertook a massive proteomics and phosphoproteomics study to define how proteins and protein phosphorylation changes throughout the meiotic cell cycle in budding yeast. They isolated protein every fifteen minutes from a prophase I arrest to ensure that they could see modest changes that occur throughout the cell cycle, without missing any of the stages. They performed two biological replicates and described the overlap between replicates. There were several interesting results from the study. First, there are a small number of proteins that change levels throughout meiosis, some of which were not previously known. Second, there are distinct waves of phosphorylation at different stages. Although this results was expected, the list of the proteins and phosphorylation sites will be very useful. They were able to put the proteins into different clusters that showed similarities in patterns. Third, they identified the potential kinases that had an enrichment of phosphorylation of specific proteins within the clusters and waves throughout meiosis. Finally, they compared wildtype and spo13 and spo13-m2 mutants that only undergo one division. They found that there are only a small number of changes in the spo13 mutants compared to wildtype which suggest that there is a delay in meiosis I events and premature meiosis II events. They also found that there was a specific set of phosphorylation events in the spo13 mutants that were missing in comparison to wildtype. The results suggest that there Cdc5 targets in meiosis I and Hrr25 targets in meiosis II that were missing in the spo13 mutants. The spo13-m2 mutants showed a more modest difference in the protein. This mutant is supposed to fail to bind Cdc5, but still may have some interactions. However, the list of overlapping potential difference between wildtype and spo13 mutant allowed them to identify another potential consensus motif for Polo kinase phosphorylation.

Overall, this study produced a massive amount of data that will likely be very useful for the yeast meiosis community to determine changes in protein phosphorylation at each meiotic stage. The comparison of wildtype with spo13 mutants is also very nice because it shows that one of the most important changes in spo13 is the lack of specific phosphorylation events. The Spo13 protein has long been of interest to the meiosis community, in that the mutants give a strong phenotype, yet the role of the protein was generally not understood. This work highlights one of the major roles of helping direct Polo kinase to specific targets. This role is also very likely to be conserved with some of the functional homologs of Spo13 in other organisms. In addition to the yeast community, these results will be of major interest to cell cycle researchers in general by increasing our understanding of how changes in phosphorylation occur throughout cell cycle stages, how differences in meiosis can allow for two divisions without an intervening S phase, and how the strategies of phosphorylation/dephosphorylation throughout the cell cycle differ between yeast, mammals, and sea star (which have proteomic studies).

The manuscript was incredibly thorough, with a lot of exciting findings. I do not have any experiments to recommend, just a couple of notes of clarification.

1) The authors use the notation of Spo13Meikin throughout, which suggests that it is a Meikin homolog. They mention that it is not a true homolog, but I think this connotation can be misleading because there may be some functions that are similar and some that are completely different. They should just call it Spo13.

- 2) In the results, I am wondering if the authors can comment on their coverage of the meiotic proteome. Do they think that the numbers represent most of the meiotic proteome (3,600-4,000 proteins, 6,700-8,700 phosphorylation sites). How does this study compare to others, like in Cheng et al, 2018?
- 3) The legend in Figure S1 should note that the time is from prophase I release.
- 4) For Figure 3D and EV4 comparing the potential kinase motif enrichment some of the bars have asterisks and some do not. In some graphs they write ns if it is not significant. But, some of the unlabeled ones look significant, like Atg1. Are these significant? And the bottom green graph, are the unlabeled ones significant or not significant?
- 5) Similarly, in Figure 4B, if it does not have an asterisk, is the difference not significant?
- 6) There is reference to Figure EV4C on page 8, but I don't think that is correct (EV4 does not have a C).
- 7) On page 10, cluster 9 seems very interesting. Can the authors comment, as they did for the others?

Referee #1:

The manuscript by Martson and colleagues describes a study where MS-based phospho-proteomics analysis was used to study phosphorylation events associated with Meiosis I and Meiosis II in budding yeast. The authors raised two general key questions, (1) how is the cell cycle machinery remodelled to direct the unique pattern of segregation in meiosis I, (2) what ensures that meiosis I is followed by another M phase, meiosis II, rather than S phase as in the canonical cell cycle? They observed distinct waves of phosphorylation signatures across the meiotic divisions and identified 1,233 significantly dynamic phosphorylation sites of which the majority (78%) showed increased abundance after prophase, while ~21% decreased. Interestingly, they found a relatively large cluster of sites with the consensus site sequence fitting the one for Polo kinase Cdc5 in Meiosis I, while the CDK sites showed a somewhat biphasic profile in MI and MII. They also found that the effect of a deletion of, or a mutation in Spo13, a member of a Meikin family of meiosis-specific master regulators influences the substrate site selection by Polo-Cdc5, with enrichment of the Polo kinase consensus motif sub-version with a phenylalanine at position +1. The Spo13Meikin binds the Polo-box of Cdc5 and the authors proposed that this interaction somehow must modulate structural features of the active site region in the Cdc5 kinase and introduce the meiosis-specific (+1F) change to the substrate consensus phosphorylation motif.

The manuscript is well-written and the study is technically well performed. The findings of temporal waves of phosphorylation events (at Polo kinase, basophilic and CDK motifs) and the Spo13Meikin-aided meiosis-specific substrate targeting are important contributions to the field. However, the claim of the discovery of a new meiosis-specific consensus sequence for Polo-Cdc5-Spo13Meikin complex is too premature and requires more experimental proof. Both biochemical and in vivo evidence are needed for the claim that Spo13Meikin "re-wires the phosphoproteome" that could explain the roles of Spo13Meikin in sister kinetochore mono-orientation, cohesion protection and the execution of two meiotic divisions.

We thank the reviewer for their appreciation of our data and analysis. We agree with the reviewer that some statements in our original manuscript were too strong and have toned down or qualified statements about the Spo13-Cdc5 consensus accordingly throughout the manuscript. We have also obtained further evidence for some conclusions, as outlined below. Although we agree it is an important avenue of future investigation, direct biochemical evidence to understand the mechanistic basis of the altered phosphoproteome of *spo13Δ* cells is beyond the scope of the current study.

Specific points:

1) The authors suggest that phosphatases active at the meiosis I to II transition could have a preference for Cdc28Cdk1 sites with lysine or arginine in the +2 position, while sites with a different residue at +2 are protected from dephosphorylation. The authors likely meant +3, not +2. However, the main point is that the evidence for this suggestion has very little experimental backing. Are the authors sure that the majority of minimal S/TP sites without the lysin or arginine, that they use as a control group, are really phosphorylated by CDK and not by other kinases with similar +1P specificity?

We apologise for the confusion, the reviewer is correct, we meant +3.

The idea that phosphatases have different specificities, and that Cdc14 phosphatase may have higher affinity for [ST]*Px[KR] vs [ST]*P sites in particular, is not novel (Bremmer *et al*, 2012) and differential dephosphorylation timing between these motifs has been observed and attributed to phosphatase specificity previously for mitotic exit based on similar data (Touati *et al*, 2018). Therefore, we believe it is important and appropriate to raise this point in the discussion (it was not mentioned in the results).

However, we agree with the reviewer's concerns that the minimal [ST]*P sites can be phosphorylated by other kinases and that this is an important point to consider. To address this, we performed a new experiment. We inhibited Cdc28^{Cdk1} specifically and acutely (using analog-sensitive *cdc28-as*) and analysed the phosphoproteome to determine the extent of decrease of phosphorylation on minimal [ST]*P sites compared to strict [ST]*Px[KR] sites in both meiosis I and meiosis II. This dataset, shown in Appendix Figures 8, 9 and Figure EV4 revealed a much greater decrease in phosphorylation on

strict [ST]*Px[KR] sites compared to [ST]*P sites upon Cdc28^{Cdk1} inhibition. This is consistent with the idea that the strict [ST]*Px[KR] motif is the preferred phosphorylation site of Cdc28^{Cdk1} inhibition and that other kinases, for example MAP kinases could phosphorylate the minimal [ST]*P at the meiosis I to II transition. This data and these points are described in a new results section starting on page 11 "[ST]*Px[KR] phosphorylation is the best predictor of Cdc28^{Cdk1} kinase activity". We also added the following text to the discussion, starting on page 18 "However, our phosphoproteomic analysis of *cdc28-as* inhibition found that [ST]*Px[KR] inhibition is the best predictor of Cdc28^{Cdk1} activity. Therefore, an alternative, more straightforward explanation is that the majority of [ST]*P sites are phosphorylated by other kinases which remain active at this transition."

2) Direct mechanistic studies are required to test these ideas and also to understand how "Cdks are permitted to rise again for entry into meiosis II". Also, replication factors for licensing have many +3K/R CDK consensus sites. How the model of drop of optimal site phosphorylation would provide maximal protection against the re-licensing of origins between MI and MII? Can the cyclin quantitative profile parallel to the optimal site drop (Fig 5F) be measured? Which cyclins are expressed and degraded?

We agree that direct mechanistic studies are required to test these ideas, but these are beyond the scope of the current study. In answer to the specific points about replication factors and the cyclin profiles, these have been studied previously. The cited study in our paper performed a thorough analysis of the RNA, protein and kinase activity attributed to Cdc28 associated with each of the meiotic cyclins and discovered interesting post-translational regulation, indicating that cyclin protein abundance does not match activity (Carlile & Amon, 2008). Previous work has also shown the involvement of multiple kinases in inhibiting DNA replication between meiosis I and II (Phizicky *et al*, 2018). We added the following text to the introduction on page 4 to make the reader aware of this point and refer them to previous work in this area. "Distinct combinations of cyclins complex with Cdc28^{Cdk1} during the meiotic divisions, however cyclin abundance does not correspond to cyclin-Cdk activity and cyclin post-translational modifications are likely to be important in defining cyclin-Cdk activity (Carlile & Amon, 2008). Both Cdk and Ime2 prevent loading of the replicative helicase at the meiosis I to II transition (Phizicky *et al*, 2018; Holt *et al*, 2007)."

We agree that these are interesting questions, which remain unresolved, but it is our hope that our study, as a resource for the field, will help inform future directed studies in these areas. To facilitate this, we have developed a web-based platform which will allow the community to display the protein abundance and phosphosite profile for an individual protein or groups of proteins across meiosis with a few clicks. <https://lori-koch.shinyapps.io/meiosis-app/>.

3) The authors suggest that Cdc5^{Polo} uses its Polo Binding Domain (PBD) to bind to phosphorylated Spo13Meikin in the same way it normally binds to substrates, and since the usual mode of substrate binding is blocked, Spo13-Cdc5^{Polo} must therefore use another mechanism to target substrates. They propose that potentially, Spo13Meikin itself targets Spo13Meikin-Cdc5^{Polo} to substrates. It is hard to imagine without any biochemical or structural data how the preference for phenylalanine (F) or glycine (G) in the +1 position can be gained via such a mechanism. Highly unbelievable and pure speculation. It could be that polo-box docking just broadens the specificity and shielding this docking via Meikin would amplify the +1F-containing optimal consensus sites. There could be other explanations. It is a too bold statement that by redirecting Cdc5^{Polo} to specific substrates containing the [DEN]x[ST]*F motif at metaphase I, Spo13Meikin establishes the meiotic programme. The subtitle of the section certainly cannot be "Identification of a Spo13Meikin - Cdc5^{Polo} consensus motif".

We agree with the reviewer that there are alternative possibilities and that understanding the mechanism requires extensive biochemical analysis which is beyond the scope of this study and have removed the strong statements referred to by the reviewer.

To try and gain further information on the substrates phosphorylated by Cdc5^{Polo} in meiosis we performed an additional phosphoproteomics experiment. We acutely inhibited Cdc5^{Polo} in meiotic cells carrying *cdc5-as* and analysed the sequence of phosphosites that were decreased in abundance in *cdc5-as* compared to wild type cells. This indicated that, indeed, as the reviewer suggests, Cdc5^{Polo} may show specificity for [DEN]x[ST]*F over sites lacking F in the +1 position. These results are shown in Figure EV6 and described in a new results section starting on page 14: "[DEN]x[ST]*F motif site phosphorylation depends on Cdc5^{Polo} activity *in vivo*".

However, we also performed an additional analysis to determine whether proteins whose phosphorylation we found to depend on Spo13 also have a PBD motif and found that the majority do. This analysis, shown in Appendix Figure 12 and described in the new results section starting on page 15 “Substrates whose phosphorylation depends on Spo13 are enriched for the polo-box binding motif S[ST]P”.

We have revised the discussion, starting page 19, to include more balanced speculation of potential mechanisms by which Spo13 influences Cdc5^{Polo}-dependent phosphorylation, making it clear that future studies are required to address the mechanism.

4) Fig 6 C, d and E, F are not clearly described. What are these sequence logos really showing? Also, how do the authors explain the effect of Spo13 deletion on CDK and Ipl1 (basophilic) kinase effects?

The IceLogo motif logos were generated using the sequences in the same categories as the pie charts above, this is indicated on the figure and described in the legend. We can only speculate the reason for the effects on the CDK consensus sites (the minimal [ST]*P motif is decreased in *spo13Δ*), however, Spo13 has been implicated in regulating Clb1-CDK activity in two recent studies (Oz *et al*, 2022; Rojas *et al*, 2023). Oz *et al* (2022) report that a metaphase I arrest caused by depletion of *CDC20* is overcome by deletion of *SPO13* or *CLB1*. while Rojas *et al* (2023) provide evidence that Spo13 promotes Clb1 phosphorylation, which in turn restricts the activity of a particular form of the APC/C. These published findings suggest that the effects on CDK consensus sites that we observe in *spo13Δ* may be a consequence of altered Clb1-CDK activity.

We also obtained further support for potential crosstalk between Cdc5 and Cdc28-Clb1 from our new Cdc5 pulldown experiment shown in Figure 6 and described on page 12. We found that Clb1 co-immunoprecipitates with Cdc5. We also identified several sites on Clb1 that showed reduced phosphorylation in *spo13Δ* cells. Therefore, while the underlying mechanisms need to be worked out, at least in principle, Cdc5 could influence Cdc28-Clb1-dependent phosphorylation and vice versa.

Although the motif recognised by basophilic kinases, which include Ipl1, was increased in the timecourse in *spo13Δ* cells (Figure 4C), we obtained no evidence that this was the case in metaphase I arrested cells. Phosphorylation on basophilic sites was enriched among the ‘no change’ sites in the comparison between wild-type and *spo13* mutants in the metaphase I arrest experiment featured in what is now Figure 7. We also found no increased phosphorylation of these basophilic sites in the *spo13* metaphase I cells (Appendix Figure S11C-D). We interpret these results to mean that any effect of *spo13Δ* on basophilic sites occurs outside of metaphase I. This is now included in the results section on page 13.

5) Page 10-11 Cluster 10 showed a peak of phosphorylation in meiosis I in wild type and a delay in phosphorylation in *spo13D* and was enriched for the strict Cdc28Cdk motif [ST]*x[KR] (Fig 5C). The +1Pro is missing?

We have corrected this mistake.

Referee #2:

This manuscript from Koch *et al* describes a mass spectrometry-based investigation of changes in the yeast phosphoproteome during meiosis. The authors isolate samples across a well-synchronized meiotic time course and at each time point analyze the abundance of both total protein and phosphopeptides. The authors identify >1000 phosphosites that are dynamic across the time course. These data were then analyzed in two ways: first, phosphosites were clustered, sorting them into 11 different patterns of behavior showing enrichment or loss at specific times in meiosis. Second, the regions around the phosphoacceptor in each peptide were compared to established consensus sequences for different kinases known to be involved in meiosis and spore formation. This analysis

correlates particular clusters with specific kinases and, therefore, implicates their activity at specific points in meiosis. It reveals, among other things, two waves of phosphorylation matching the CDK consensus site as well as two waves matching the Cdc5 consensus.

The authors then apply the same protocol to analyze *spo13* mutant cells and compare them to the wild-type. Spo13 is known to bind to the Cdc5 kinase and is required for cells to undergo two successive chromosome divisions during meiosis. This analysis revealed about three hundred phosphosites whose abundance was altered in the *spo13* mutant cells. Applying the consensus motifs to these sites revealed not only that the Cdc5 consensus was over-represented in the set of decreased phosphosites, but that other kinase consensus sites were also affected, suggesting that these kinases are regulated by Spo13.

Finally the authors specifically compared phosphopeptides from WT, *spo13Δ* and *spo13-m2* (an allele that does not bind Cdc5) cells arrested at metaphase I, when Spo13 is thought to be most active. The phosphosites reduced in these conditions reveal not just a reduction in Cdc5 consensus sites, but specifically a slightly modified Cdc5 consensus, suggesting that this represents a favored site of Spo13-Cdc5 phosphorylation.

This is a very thorough study that generates significant new hypotheses about the regulator wiring of meiotic kinases. The timing of the increase or decrease of phosphorylation on consensus site of known kinases largely matches expectations and this gives confidence that the newer insights are correct.

A weakness of this study is that the analysis is based entirely on consensus site and motif frequency. Many different kinases fall into the category of acid/base/proline directed and so the appearance of a particular phosphosite within a consensus can't be taken as firm evidence that the phosphorylation is performed by a particular kinase. In some cases, the conclusions are weaker than others. In particular, the inference that Spo13 regulates Hrr25 seems quite speculative and should be qualified in the text.

We agree with the reviewer that consensus site phosphorylation alone cannot be taken as evidence of phosphorylation by a particular kinase and we have made it clear throughout the text what is speculation. In addition, to provide more confidence in the consensus motifs for Cdc5^{Polo} and Cdc28^{Cdk1}, we generated two additional phosphoproteomic datasets after acute inhibition of these kinases. This analysis suggested that the sites that depend most on Cdc5^{Polo} and Cdc28^{Cdk1} during meiosis are respectively [DEN]x[ST]*F and [ST]*Px[KR].

The work would have been strengthened by direct biochemical demonstration of the connection between some of the sites and particular kinases. For example, testing whether Spo13-Cdc5 has a different phosphosite preference than Cdc5 alone. However, this is a general issue with all 'omic level analyses and, this issue notwithstanding, the study is very well done and generates not only a large body of information that will be useful to the field, but also specific models and hypotheses for further, mechanistic analysis.

We appreciate the reviewer's support of this study. We agree that biochemical analysis of Spo13-Cdc5^{Polo} would be extremely interesting, however, initial attempts have indicated that this is a very challenging project which is beyond the scope of the current study.

Other comments:

In many cases, eg. the graphs for clusters 1, 2, and 4 in Figure 5B, there does not appear to be any difference between the median intensity curves for WT and *spo13Δ*. This seems surprising given that the clusters have been formed from sites that "significantly vary" between the two strains.

This is a technical point. The criterion for those that "significantly vary" is a significant difference at one time point, while these lineplots show the median over the entire the time-course of all the sites in that cluster. Where the median intensity curves look similar it means that the 'significant difference' between wild type and *spo13Δ* was not always at the same timepoint.

It would be useful in the supplementary data to indicate which phosphosites fell into which consensus

so that readers could see which sites on which proteins are implicated as specific targets of the different kinases.

Thank you for this excellent suggestion. To ensure our data is as useful for the community as possible, we have generated an interactive website that allows proteins, their phosphorylation and consensus motifs to be mined.

Minor points

Figure EV5 legend "G" should be "F"

We thank the reviewer for noticing this and will correct it (now Figure EV3).

Referee #3:

Koch et al undertook a massive proteomics and phosphoproteomics study to define how proteins and protein phosphorylation changes throughout the meiotic cell cycle in budding yeast. They isolated protein every fifteen minutes from a prophase I arrest to ensure that they could see modest changes that occur throughout the cell cycle, without missing any of the stages. They performed two biological replicates and described the overlap between replicates. There were several interesting results from the study. First, there are a small number of proteins that change levels throughout meiosis, some of which were not previously known. Second, there are distinct waves of phosphorylation at different stages. Although this results was expected, the list of the proteins and phosphorylation sites will be very useful. They were able to put the proteins into different clusters that showed similarities in patterns. Third, they identified the potential kinases that had an enrichment of phosphorylation of specific proteins within the clusters and waves throughout meiosis. Finally, they compared wildtype and spo13 and spo13-m2 mutants that only undergo one division. They found that there are only a small number of changes in the spo13 mutants compared to wildtype which suggest that there is a delay in meiosis I events and premature meiosis II events. They also found that there was a specific set of phosphorylation events in the spo13 mutants that were missing in comparison to wildtype. The results suggest that there Cdc5 targets in meiosis I and Hrr25 targets in meiosis II that were missing in the spo13 mutants. The spo13-m2 mutants showed a more modest difference in the protein. This mutant is supposed to fail to bind Cdc5, but still may have some interactions. However, the list of overlapping potential difference between wildtype and spo13 mutant allowed them to identify another potential consensus motif for Polo kinase phosphorylation.

Overall, this study produced a massive amount of data that will likely be very useful for the yeast meiosis community to determine changes in protein phosphorylation at each meiotic stage. The comparison of wildtype with spo13 mutants is also very nice because it shows that one of the most important changes in spo13 is the lack of specific phosphorylation events. The Spo13 protein has long been of interest to the meiosis community, in that the mutants give a strong phenotype, yet the role of the protein was generally not understood. This work highlights one of the major roles of helping direct Polo kinase to specific targets. This role is also very likely to be conserved with some of the functional homologs of Spo13 in other organisms. In addition to the yeast community, these results will be of major interest to cell cycle researchers in general by increasing our understanding of how changes in phosphorylation occur throughout cell cycle stages, how differences in meiosis can allow for two divisions without an intervening S phase, and how the strategies of phosphorylation/dephosphorylation throughout the cell cycle differ between yeast, mammals, and sea star (which have proteomic studies).

The manuscript was incredibly thorough, with a lot of exciting findings. I do not have any experiments to recommend, just a couple of notes of clarification.

1) The authors use the notation of Spo13Meikin throughout, which suggests that it is a Meikin homolog. They mention that it is not a true homolog, but I think this connotation can be misleading because there may be some functions that are similar and some that are completely different. They should just call it Spo13.

We agree with the reviewer, this is an important point. We originally used Spo13^{Meikin} because of a recent EMBO J paper that had used this annotation and made the strong argument that Spo13 is a

Meikin homolog. We have changed the notation throughout and now call it Spo13 as the reviewer suggests.

2) In the results, I am wondering if the authors can comment on their coverage of the meiotic proteome. Do they think that the numbers represent most of the meiotic proteome (3,600-4,000 proteins, 6,700-8,700 phosphorylation sites). How does this study compare to others, like in Cheng et al, 2018?

We have added a sentence to the first results section on page 5 citing recent similar yeast proteomic studies. "The number of proteins and phospho-sites quantified in our experiments is comparable to other recent proteomic studies (Paulo *et al*, 2015; Cheng *et al*, 2018; Li *et al*, 2019)." We used strict criteria for inclusion in the analysed phosphoproteomics dataset. We required detection in multiple conditions and protein information for normalisation, which is not always standard in the field. This decision to prioritise high quality over volume, combined with technical limitations of the method means that our data likely do not represent the full extent of phosphorylation in meiosis, however all identified sites are high confidence.

3) The legend in Figure S1 should note that the time is from prophase I release.

We have added this.

4) For Figure 3D and EV4 comparing the potential kinase motif enrichment some of the bars have asterisks and some do not. In some graphs they write ns if it is not significant. But, some of the unlabeled ones look significant, like Atg1. Are these significant? And the bottom green graph, are the unlabeled ones significant or not significant?

Everything unlabelled is not significant based on Fisher test, we have added this point to the methods. The lack of significance in many cases is due to the size of the groups which is often small.

5) Similarly, in Figure 4B, if it does not have an asterisk, is the difference not significant?

It is not significant, as described in the response to point 4.

6) There is reference to Figure EV4C on page 8, but I don't think that is correct (EV4 does not have a C).

This has been corrected.

7) On page 10, cluster 9 seems very interesting. Can the authors comment, as they did for the others?

We added the following text "Finally, we note acidic residues at -3 and -2 of the cluster 9 consensus, suggesting potential phosphorylation by Cdc5^{Polo} or Hrr25^{CK1} (Fig 5F). Phosphorylation of Hrr25 itself on serine 330, which has been documented in mitotic cells (Breitkreutz *et al*, 2010; Zhou *et al*, 2021) was found in this cluster (Fig 5F). Cdc5^{Polo} and Hrr25^{CK1} physically interact (Galander *et al*, 2019), raising the possibility that Hrr25-S330 is regulated by Cdc5^{Polo} bound to Spo13."

References

Breitkreutz A, Choi H, Sharom JR, Boucher L, Neduva V, Larsen B, Lin ZY, Breitkreutz BJ, Stark C, Liu G, *et al* (2010) A global protein kinase and phosphatase interaction network in yeast. *Science (New York, NY)* 328: 1043–1046

Bremmer SC, Hall H, Martinez JS, Eissler CL, Hinrichsen TH, Rossie S, Parker LL, Hall MC & Charbonneau H (2012) Cdc14 phosphatases preferentially dephosphorylate a subset of cyclin-dependent kinase (Cdk) sites containing phosphoserine. *J Biol Chem* 287: 1662–1669

Carlile TM & Amon A (2008) Meiosis I is established through division-specific translational control of a cyclin. *Cell* 133: 280–291

- Cheng Z, Otto GM, Powers EN, Keskin A, Mertins P, Carr SA, Jovanovic M & Brar GA (2018) Pervasive, Coordinated Protein-Level Changes Driven by Transcript Isoform Switching during Meiosis. *Cell* 172: 910--923.e16
- Galander S, Barton RE, Borek WE, Spanos C, Kelly DA, Robertson D, Rappsilber J & Marston AL (2019) Reductional Meiosis I Chromosome Segregation Is Established by Coordination of Key Meiotic Kinases. *Developmental Cell* 49: 526-541.e5
- Holt LJ, Hutti JE, Cantley LC & Morgan DO (2007) Evolution of Ime2 Phosphorylation Sites on Cdk1 Substrates Provides a Mechanism to Limit the Effects of the Phosphatase Cdc14 in Meiosis. *Molecular Cell* 25: 689--702
- Li J, Paulo JA, Nusinow DP, Huttlin EL & Gygi SP (2019) Investigation of Proteomic and Phosphoproteomic Responses to Signaling Network Perturbations Reveals Functional Pathway Organizations in Yeast. *Cell Rep* 29: 2092-2104.e4
- Oz T, Mengoli V, Rojas J, Jonak K, Braun M, Zagoriy I & Zachariae W (2022) The Spo13/Meikin pathway confines the onset of gamete differentiation to meiosis II in yeast. *The EMBO Journal* 41: e109446
- Paulo JA, O'Connell JD, Gaun A & Gygi SP (2015) Proteome-wide quantitative multiplexed profiling of protein expression: carbon-source dependency in *Saccharomyces cerevisiae*. *Mol Biol Cell* 26: 4063--4074
- Phizicky DV, Berchowitz LE & Bell SP (2018) Multiple kinases inhibit origin licensing and helicase activation to ensure reductive cell division during meiosis. *eLife* 7: e33309
- Rojas J, Oz T, Jonak K, Lyzak O, Massaad V, Biriuk O & Zachariae W (2023) Spo13/MEIKIN ensures a Two-Division meiosis by preventing the activation of APC/C_{Ama1} at meiosis I. *EMBO J* 42: e114288
- Touati SA, Kataria M, Jones AW, Snijders AP & Uhlmann F (2018) Phosphoproteome dynamics during mitotic exit in budding yeast. *Embo J* 37: e98745
- Zhou X, Li W, Liu Y & Amon A (2021) Cross-compartment signal propagation in the mitotic exit network. *Elife* 10: e63645

Prof. Adele L Marston
University of Edinburgh
Wellcome Centre for Cell Biology
Max Born Crescent
Edinburgh EH9 3BF
United Kingdom

31st Jan 2024

Re: EMBOJ-2023-115604R
Rewiring of the phosphoproteome executes two meiotic divisions

Dear Adele,

Thank you for submitting your revised manuscript to The EMBO Journal. Referee 1 has now looked at it once again, and I am happy to say was fully satisfied with the revisions. We should therefore be able to swiftly proceed with its acceptance and publication, as soon as the following editorial points have been incorporated:

1) Figure legend/data analysis issues raised by our data editors' pre-acceptance checks:

- A separate 'Data Information' section is required in the legends of figures EV 4a, d, f.
- Please indicate the statistical test used for data analysis in the legends of figures 3b; 6a-b, d; 8a-b; EV 1c-d; EV 5c.
- Please note that in figures 3e; 4b-c; 7c-f; 8e; EV 6d; there is a mismatch between the annotated p values in the figure legend and the annotated p values in the figure file that should be corrected.
- Please note that information related to n is missing in the legends of figures 6a-e.
- Please note that the measure of center for the error bars needs to be defined in the legends of figures 6c, e.

2) Related to the main manuscript text:

- Please consider a somewhat less terse, more explicit title. E.g., "Pervasive phosphoproteome changes/rewiring drive(s) successive meiotic divisions in budding yeast"
- Please correct the header of the competing interest section to "Disclosure and competing interests statement"
- As we are switching from a free-text author contribution statement towards a more formal statement based on Contributor Role Taxonomy (CRediT) terms, please remove the present Author Contribution section and instead specify each author's contribution(s) directly in the Author Information page of our submission system during upload of the final manuscript. See <https://casrai.org/credit/> for more information.
- Please double-check to make sure to all relevant funding information in the manuscript is congruent with the info entered into our submission system. (Missing in the system currently: Wellcome Multi-User Equipment Grant [091020]).
- Please rename the Methods section to Material and Methods.
- Please remove review access information, and ensure that data listed in the Data Availability section becomes publicly accessible at this point, latest upon formal acceptance.
- Please rename the current "Table EV1" into "Table 1", as it seems it should be regularly included in the main manuscript.

3) Issues with the Appendix PDF:

- Please convert all the "Appendix Tables" into EXPANDED VIEW DATASETS and upload them separately as one spreadsheet file for each of them. They should be called (and referenced as) "Dataset EV1/2/3...", and have their title and legend each on a separate tab of the XLSX file, instead of in the Appendix.
- Similarly, reference to the R code should be removed from the Appendix (it should be referenced in the Material and Methods section though), and the respective file should not be uploaded as Source Data, but as EXPANDED VIEW COMPUTER CODE in a ZIP file - nomenclature/in-text call-out "ComputerCode EV1"
- Please add page numbers in the Table of Contents

4) Source Data files:

- Please combine the source data files for the EXPANDED VIEW FIGURES in one single ZIP archive before re-uploading (and also the Appendix Source Data should be combined in one specific archive). The main figure source data are already correctly uploaded as one archive file per figure, but please double check their completeness - the Source Data checklist lists a Fig 6F that seems to be missing, please specify if/how it may have been renamed?

5) Synopsis material:

- Please provide suggestions for a short 'blurb' text prefacing and summing up the study in two sentences (max. 250 characters), followed by 3-5 one-sentence 'bullet points' with brief factual statements about key results of the paper; they will form the basis of an editor-written 'Synopsis' accompanying the online version of the article (see new articles on our journal website for some recent examples). Please also provide a simple synopsis image, which can be used as a "visual title" for the synopsis section of

your paper. The image should be in PNG or JPG format with the modest dimensions of exactly 550 pixels wide x 300-600 pixels high.

I am therefore returning the manuscript to you for a final round of minor revision, only to allow you to make these adjustments and upload all modified files. Let me know if anything should be unclear in this regard!

With kind regards,
Hartmut

9) Digital image enhancement is acceptable practice, as long as it accurately represents the original data and conforms to community standards. If a figure has been subjected to significant electronic manipulation, this must be clearly noted in the figure legend and/or the 'Materials and Methods' section. The editors reserve the right to request original versions of figures and the original images that were used to assemble the figure. Finally, we generally encourage uploading of numerical as well as gel/blot image source data; for details see: embopress.org/page/journal/14602075/authorguide#sourcedata

At EMBO Press, we ask authors to provide source data for the main manuscript figures. Our source data coordinator will contact you to discuss which figure panels we would need source data for and will also provide you with helpful tips on how to upload and organize the files.

Further information is available in our Guide For Authors:

In the interest of ensuring the conceptual advance provided by the work, we recommend submitting a revision within 3 months (30th Apr 2024). Please discuss the revision progress ahead of this time with the editor if you require more time to complete the revisions. Use the link below to submit your revision:

Link Not Available

Referee #1:

The authors have sufficiently addressed the points raised. I recommend the manuscript for publishing in EMBO J.

Prof. Adele L Marston
University of Edinburgh
Wellcome Centre for Cell Biology
Max Born Crescent
Edinburgh EH9 3BF
United Kingdom

7th Feb 2024

Re: EMBOJ-2023-115604R1
Rewiring of the phosphoproteome executes two meiotic divisions in budding yeast

Dear Adele,

Thank you for submitting your final revised manuscript for our consideration. I am pleased to inform you that we have now accepted it for publication in The EMBO Journal.

With kind regards,

Hartmut
